# KIT ligand protects against both light-induced and genetic photoreceptor degeneration

Huirong Li[1,2], Lili Lian[1], Bo Liu[1], Yu Chen[1,2], Jinglei Yang[2], Shuhui Jian[1], Jiajia Zhou[1], Ying Xu[3], Xiaoyin Ma[1,2], Jia Qu[2]*, Ling Hou[1,2]*

[1]Laboratory of Developmental Cell Biology and Disease, School of Ophthalmology and Optometry and Eye Hospital, Wenzhou Medical University, Wenzhou, China; [2]State Key Laboratory of Ophthalmology, Optometry and Vision Science, Wenzhou Medical University, Wenzhou, China; [3]GHM Institute of CNS Regeneration, Jinan University, Guangzhou, China

**Abstract** Photoreceptor degeneration is a major cause of blindness and a considerable health burden during aging but effective therapeutic or preventive strategies have not so far become readily available. Here, we show in mouse models that signaling through the tyrosine kinase receptor KIT protects photoreceptor cells against both light-induced and inherited retinal degeneration. Upon light damage, photoreceptor cells upregulate Kit ligand (KITL) and activate KIT signaling, which in turn induces nuclear accumulation of the transcription factor NRF2 and stimulates the expression of the antioxidant gene *Hmox1*. Conversely, a viable *Kit* mutation promotes light-induced photoreceptor damage, which is reversed by experimental expression of *Hmox1*. Furthermore, overexpression of KITL from a viral AAV8 vector prevents photoreceptor cell death and partially restores retinal function after light damage or in genetic model**s** of human retinitis pigmentosa. Hence, application of KITL may provide a novel therapeutic avenue for prevention or treatment of retinal degenerative diseases.

*For correspondence:
jia.qu@163.com (JQ);
lhou@eye.ac.cn (LH)

Competing interests: The authors declare that no competing interests exist.

## Introduction

Retinal degeneration due to photoreceptor loss is a major threat to human health as progressive vision loss severely interferes with a person's daily activities. Such photoreceptor cell loss is common to a number of degenerative eye disorders including cone dystrophy, retinitis pigmentosa (RP) and the highly prevalent age-related macular degeneration (AMD) (*Ma et al., 2019*; *Mitchell et al., 2018*; *Wright et al., 2010*). Each of these disorders is influenced, to various degrees, by mutations in a number of distinct genes. RP, for instance, has been associated with mutations in about 80 genes and can be inherited as an autosomal-dominant, autosomal-recessive, or X-linked trait (*Dias et al., 2018*). It is characterized initially by degeneration of rod photoreceptors, followed by a loss of cone photoreceptors and of the photoreceptor-trophic retinal pigment epithelium (RPE). Consequently, patients suffer from progressive night blindness, tunnel vision and, rarely, may become totally blind (*Dias et al., 2018*; *Hartong et al., 2006*).

Currently, one of the major challenges for the field of retinal degenerative diseases is to provide strategies to prevent or delay the onset of photoreceptor cell loss. Several approaches are currently under investigation to treat vision loss in patients suffering from retinal degenerations, including retinal prostheses implants, stem cell transplants and gene therapy (*Gagliardi et al., 2019*; *Scholl et al., 2016*). Although gene therapy has recently been approved for a specific causative mutation (RPE65) (*Russell et al., 2017*), there are still no broadly applicable and efficacious treatments available to prevent or cure vision loss caused by photoreceptor degeneration. Alternatively,

photoreceptor cell death could be prevented by strengthening endogenous prosurvival mechanisms or by directly blocking cell death (*Pardue and Allen, 2018*; *Trifunović et al., 2012*). In fact, several neuroprotective factors have been identified based on results obtained with transgenic and conditional knockout animals and pharmacological treatments (*Joly et al., 2008*; *Ueki et al., 2009*). They include ciliary neurotrophic factor (CNTF), glial cell–derived neurotrophic factor (GDNF), pigment epithelium-derived factor (PEDF) and rod-derived cone viability factor (RdCVF), all of which might potentially be used to treat photoreceptor degeneration in humans (*Aït-Ali et al., 2015*; *Chen et al., 2019*; *Del Río et al., 2011*; *Wen et al., 2012*). Their clinical application, however, is hampered by relatively short half-lives, systemic side effects and reduced efficacy, and none are commercially available for clinical use (*Pardue and Allen, 2018*; *Scholl et al., 2016*). Therefore, there is a need to identify additional factors that are capable of improving photoreceptor survival and that may have pharmacokinetic properties that differ from those already studied.

Retinal degeneration, regardless of whether it is due to intrinsic genetic or extrinsic environmental conditions, is normally associated with an induction of damaging effector molecules. As a corollary of damage, the eye may respond with the induction of damage-limiting, endogenous neuroprotective factors such as the ones mentioned above (*Goldman, 2014*; *Pearson and Ali, 2018*). This fact prompted us to search for additional neuroprotective factors after experimental light damage (LD). It is known that limited light stress activates the FGF2/gp130 signaling pathway (*Liu et al., 1998*; *Ueki et al., 2009*) and that prolonged exposure to light upregulates CNTF and FGF2, although usually at levels too low to protect against retinal degeneration (*LaVail et al., 1998*; *Wen et al., 1998*). That LD is a valid strategy to identify novel retinoprotective factors is underscored by the role excessive light may play in worsening genetically influenced retinal degenerations such as AMD (*Organisciak and Vaughan, 2010*; *Suzuki et al., 2012*).

To discover novel retinoprotective factors, we applied an RNA-seq analysis to mouse neural retinas after prolonged LD. This approach confirmed the induction of a number of genes previously found to be altered during LD (*Hadziahmetovic et al., 2012*; *Rattner and Nathans, 2005*; *Rutar et al., 2015*) and identified a number of novel genes not previously recognized in this context. One of them is *Kit ligand* (*Kitl*) whose protein product, KITL (also known as stem cell factor, SCF, or mast cell growth factor, MGF), stimulates an important signaling pathway by interacting with its receptor, KIT (also known as the proto-oncogene c-KIT) and promotes cell survival, migration and proliferation (*Hou and Pavan, 2008*; *Lennartsson and Rönnstrand, 2012*; *Li and Hou, 2018*). KIT is a multi-domain transmembrane tyrosine kinase expressed in a variety of cell types including melanocytes, germ cells and cells of the hematopoietic lineage, and is required for their normal development and function (*Blume-Jensen et al., 2000*; *Comazzetto et al., 2019*; *Hou et al., 2000*; *Yuzawa et al., 2007*). Interaction of KIT with its ligand leads to receptor dimerization and autophosphorylation which in turn activates the MAPK pathway, phosphatidylinositol 3'-kinase (PI3K), JAK/STAT and SRC family members (*Hou and Pavan, 2008*; *Lennartsson and Rönnstrand, 2012*). In mice, loss-of-function mutations in *Kit* cause severe anemia, pigmentation abnormalities, sterility, mast cell deficits, spatial learning memory deficits and defects in peripheral nerve regeneration (*Gore et al., 2008*; *Motro et al., 1996*; *Wen et al., 2010*; *Zsebo et al., 1990*). Nevertheless, although KIT has been found to be expressed in retinal progenitor cells (*Koso et al., 2007*; *Zou et al., 2019*), its functional role in the adult retina is still unknown.

Here, we find that mice homozygous for the *Kit^{Wps}* mutation (*Guo et al., 2010*) show an exacerbated photoreceptor degeneration and that overexpression of KITL can prevent photoreceptor cell death in light-damaged mice. Furthermore, we show that KIT signaling stimulates the expression of *Hmox1* in an NRF2-dependent manner and that experimental expression of *Hmox1* in *Kit^{Wps}* homozygotes prevents light-induced photoreceptor degeneration. Furthermore, we show that overexpression of KITL prevents photoreceptor cell death and partially rescues the retinal dysfunction in mouse genetic models of retinitis pigmentosa. Hence, our findings suggest a mechanism by which KITL/KIT signaling contributes to protection of photoreceptor cells from degeneration and which may potentially lead to novel therapeutic strategies in retinal degenerative disorders.

## Results

### Light damage upregulates endogenous KITL in photoreceptor cells

Previous results have shown that light stress induces endogenous factors in the eye that are capable of protecting photoreceptor cells (*Liu et al., 1998*; *Ueki et al., 2009*). Hence, we used LD in light sensitive BALB/c albino mice to search for such inducible factors. Retinal damage is usually accompanied by reactive gliosis, characterized by the accumulation of filamentous proteins such as glial fibrillary acidic protein (GFAP) and formation of a glial scar (*Dyer and Cepko, 2000*). As shown in *Figure 1—figure supplement 1*, after 1 day of continuous exposure (15,000 Lux), BALB/c neural retinas showed no obvious signs of retinal degeneration, but increased expression of GFAP. After three days of continuous exposure, however, there was significant retinal degeneration (*Figure 1—figure supplement 1A to C*). Therefore, we used the 1-day exposure to identify novel protective genes (*Figure 1A*) and longer exposures to test for effects on late-stage damage. Whole neural retinas were subjected to RNA-seq analysis. The RNA-seq data showed a large number of genes whose expression was significantly changed ($p$ adjust value <0.01 and fold change >2) in the LD-treated neural retinas compared with the non-treated ones (*Figure 1A*). These genes included known light-inducible ones such as *Gfap, Edn2, Fgf2, Ccl3* and *Ccl12* (*Figure 1B*) and hence supported the validity of this approach (*Hadziahmetovic et al., 2012*; *Rattner and Nathans, 2005*; *Rutar et al., 2015*). A KEGG (Kyoto Encyclopedia of Genes and Genomes) analysis identified the MAPK pathway among the top 20 altered signaling pathways (*Figure 1C*), with increased expression, for instance, of the genes encoding LIF, KITL and NGF (*Figure 1D*). RT-PCR confirmed their differential expression identified by the RNA-seq analysis. In particular, *Kitl* expression was significantly increased in the neural retina after LD (*Figure 1—figure supplement 1D*). Because KIT signaling plays vital roles in a variety of cell types, we focused on this pathway in the retina. Quantitation of western blot bands showed that both mature transmembrane and cleaved soluble forms of KITL (45 and 19 kDa, respectively) were increased (m-KITL, 2.4 ± 0.9 fold; s-KITL, 4.5 ± 1.6 fold; n = 5) (*Figure 1E,F*). Phosphorylated KIT (indicative of activated KIT), although not total KIT, was also increased (3.14 ± 0.94, n = 5) (*Figure 1G,H*). Immunostaining showed that upregulation of KITL was predominantly seen in photoreceptor inner segments (*Figure 1I*). These results suggest that following prolonged light exposure, upregulation of KITL leads to activation of KIT signaling.

The above immunostaining data showed that KITL expression was hardly detectable in RPE cells under normal or even LD condition (*Figure 1I*). Nevertheless, given that the retinal pigment epithelium (RPE) secrets neurotrophic factors to support photoreceptor cells, we examined in more detail whether the RPE might contribute to KITL expression under normal or LD conditions. To this end, we separately isolated the RPE and neural retina as shown in *Figure 1—figure supplement 2A*. Quantitative PCR data showed higher *Kitl* expression in the neural retina (14.6 ± 7.2 fold, n = 3) compared to RPE (*Figure 1—figure supplement 2B*). Furthermore, upon LD, *Kitl* was upregulated in neural retinas (6.7 ± 1.2 fold compared to normal light condition, n = 3) but hardly in RPE (1.24 ± 0.36 fold compared to normal light condition, n = 3) (*Figure 1—figure supplement 2C,D*). To test for protein expression, we further dissected the neural retina to yield photoreceptor tissue as schematically shown in *Figure 1—figure supplement 3A*. Western blot data showed that under normal light conditions, the level of KITL protein was 5.11 ± 0.29 fold higher in the photoreceptor tissue compared to RPE (based on three independent samples) (*Figure 1—figure supplement 3B, C*). These results were supported by in situ hybridization (ISH), showing that light stress significantly increased the *Kitl* signal in photoreceptor cells but not in RPE cells (*Figure 1—figure supplement 3D*). Hence, the data suggest that photoreceptor cells rather than RPE cells are the major source of increased *Kitl* expression following light stress.

### The *Kit^Wps* mutation does not cause detectable changes in retinal function and structure but disrupts KIT activation in the retina

To determine whether upregulation of KIT signaling has any functional relevance during excessive light exposure, we used mice harboring a mutant *Kit* allele on a C57BL/6J background, *Kit^Wps*. This allele encodes a KIT protein with an Asp-to-Asn (D-to-N) mutation at residue 60 in the extracellular domain that reduces, but does not completely eliminate, KIT signaling. Consequently, *Kit^Wps* homozygotes are viable, although they are totally white and infertile (*Guo et al., 2010*). Nevertheless,

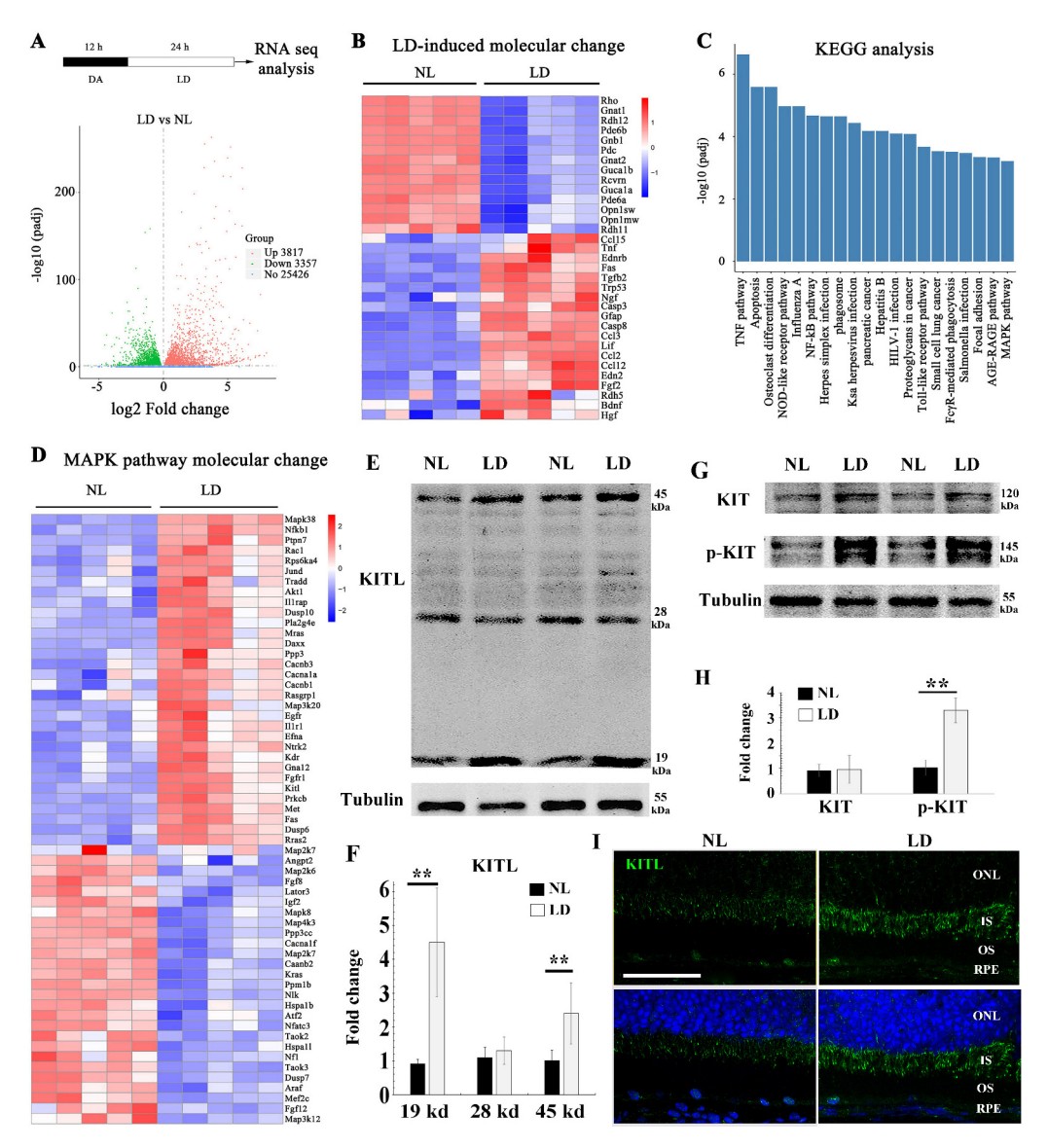

**Figure 1.** Light damage induces increased expression of KITL and activation of KIT in mouse retina. (**A**) Schematic representation of time frame and analysis of light damage (LD). Two-month-old albino mice were raised in the dark for 12 hr and then exposed to constant white light of 15,000 lux for 1 day to induce retinal damage. The volcano map of transcriptome analysis shows a global view of gene expression. (**B**) Heat map of the selective LD-induced differentially expressed reported genes. The columns for NL (normal light) or LD represent the results from five biological replicates. (**C**) The bar graph of KEGG analysis shows the top 20 differentially activated signaling pathways. (**D**) Heat map shows the differentially expressed genes of the MAPK pathway. (**E**) Western blotting analysis of KITL in retinas after LD. (**F**) Quantification of western blot bands shows the expression levels of the KITL. Note that both the 19 kDa and 45 kDa isoforms of KITL were upregulated by LD. (**G**) Western blot analysis of the tyrosinase kinase receptor KIT (upper panels) and its phosphorylated form (lower panels) in the retinas after LD. (**H**) Quantification of western blot bands show the expression levels of KIT and its phosphorylation levels. Note that p-KIT was upregulated by LD. (**I**) Immunostaining of light-treated retina detected by anti-KITL antibody. Each image is representative of at least five retinas. IS, photoreceptor inner segments; ONL, outer nuclear layer; OS, photoreceptor outer segments. ** indicates p<0.01. Scale bar, 50 μm.

The online version of this article includes the following source data and figure supplement(s) for figure 1:

**Figure supplement 1.** High-intensity light induces retinal damage in albino mice.

**Figure supplement 1—source data 1.** Source data for the diagram in Figure 1-figure supplement 1B.

**Figure supplement 2.** The level of KITL protein is much higher in photoreceptor cells compared to RPE cells.

**Figure supplement 3.** LD induces *Kitl* expression in photoreceptor cells.

$Kit^{Wps}$ homozygotes have no detectable changes in fundus, such as blood vessels and the pigmentation, similar to other point mutations in $Kit$ (*Aoki et al., 2009*; *Figure 2A*). Moreover, the amplitudes of the a- and b-waves under scotopic or photopic conditions are not significantly different between $Kit^{+/+}$ and $Kit^{Wps/Wps}$ mice (*Figure 2B,C*), and $Kit^{Wps/Wps}$ retinas appear structurally intact (*Figure 2D*). Also, the numbers of cell nuclei in the ganglion cell layer (GCL), inner nuclear layer (INL) and outer nuclear layer (ONL) were similar to those in $Kit^{+/+}$ retinas (*Figure 2E*) and immunoreactivities for markers for bipolar cells (OTX2), 'On' bipolar cells (PKCα), rod photoreceptor cells (Rhodopsin) and cone photoreceptor cells (Opsin) were as in $Kit^{+/+}$ retinas (*Figure 2F*). Likewise, there was also no significant difference in the number of bipolar cells and cone photoreceptor cells (*Figure 2F, G*). These data show that under normal conditions, the $Kit^{Wps}$ mutation does not cause detectable changes in retinal function, histology, or structure.

We next assessed whether the mutation would interfere with KIT signaling in the retina. As shown in *Figure 2—figure supplement 1*, western blots and immunostaining using the AB5506 antibody (recognizing the intracellular kinase domain of KIT) did not detect differences in KIT expression levels between $Kit^{+/+}$ and $Kit^{Wps}$ homozygous retinas. However, immunostaining using the ACK45 antibody (recognizing the extracellular immunoglobulin domain of wild type, although not $Wps/Wps$ KIT

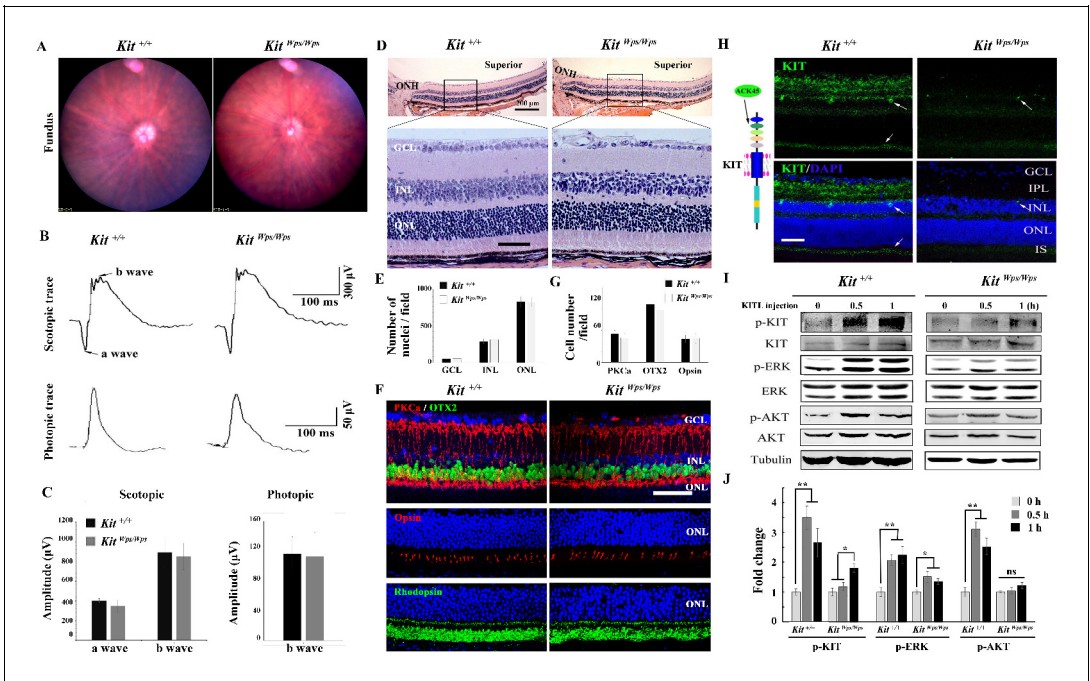

**Figure 2.** No changes in retinal cell distribution and structure, but disruption of Kit activation in $Kit^{Wps/Wps}$ retina. (A) Fundus images of 2-month-old $Kit^{+/+}$ and $Kit^{Wps/Wps}$ mice. (B) ERG traces of $Kit^{+/+}$ (left panels) and $Kit^{Wps/Wps}$ (right panels) retinas under scotopic (upper panels) and photopic conditions (lower panels). (C) Quantification of ERG amplitudes under scotopic and photopic conditions. (D, E) Histological images of H and E staining from 2-month-old $Kit^{+/+}$ and $Kit^{Wps/Wps}$ retinas (D) and quantification showing the mean number of nuclei localized at GCL, INL, and ONL over the length of 0.35 mm between 300 µm to 700 µm from the optic nerve head in dorsal retina of $Kit^{+/+}$ and $Kit^{Wps/Wps}$ mice (n = 5) (E). Representative histological sections are from superior retinas 0.3 mm from the optic nerve head. (F) Immunohistochemistry for OTX2, PKCα, Opsin and Rhodopsin in 2-month-old $Kit^{+/+}$ and $Kit^{Wps/Wps}$ retinas. (G) Quantification of OTX2+, PKCα+ and Opsin+ cells in the indicated retinas. Each image is representative for at least five retinas. The analyzed sectional numbers are from at least 30 sections from at least five retinas. (H) Schematic representation of KIT protein structure and the location recognized by the ACK45 antibody (left panels) and immunostaining images of ACK45 antibody in 3-month-old $Kit^{+/+}$ and $Kit^{Wps/Wps}$ retinas (right panels). (I) Western blots show phosphorylation of KIT, ERK and AKT in $Kit^{+/+}$ and $Kit^{Wps/Wps}$ retinas after the injection of KITL (5.6 nM) at the indicated time points. (J) Quantification of western blot bands shows the phosphorylation levels of KIT, ERK and AKT. GCL, ganglion cell layer; IPL, inner plexiform layer; INL, inner nuclear layer; IS, photoreceptor inner segments; ONL, outer nuclear layer; OS, photoreceptor outer segments. Arrows in panels H indicate KIT-positive signal. * or ** indicates p<0.05 or p<0.01. Scale bar, 50 µm.

The online version of this article includes the following source data and figure supplement(s) for figure 2:

**Source data 1.** Source data for the graphs in Figure 2C.
**Figure supplement 1.** The $Kit^{Wps}$ mutation does not affect KIT expression.
**Figure supplement 2.** KIT is expressed in photoreceptor cells.

protein) showed a significantly decreased immunoreactivity in the photoreceptor layer, the inner plexiform layer and the ganglion cell layer of $Kit^{Wps/Wps}$ retinas (*Figure 2H*). It is unlikely that the KIT immunoreactivity in the photoreceptor layer stems from interdigitating apical microvilli of Müller glial cells as double immunostaining with ACK45 and an antibody against the Müller glia-specific EAAT1 protein in 2-month-old C57BL/6J mice did not show any overlapping signals. Furthermore, the KIT signal was drastically reduced in mice homozygous for a retinal degeneration allele (*rd10/rd10*) which almost totally lack photoreceptor cells but not Müller glia cells (*Figure 2—figure supplement 2*).

As the *Wps* mutation lies in the first immunoglobulin domain of KIT, which is required for the interaction with KITL, it was likely that it would impair KIT activation by KITL not only in extraocular cells but also in the retina. Indeed, upon intravitreal injection of KITL, phosphorylation of KIT and its downstream targets ERK and AKT were prominent in $Kit^{+/+}$ but not in $Kit^{Wps/Wps}$ retinas (*Figure 2I, J*). These results indicate that the mutation substantially reduced activation of KIT by KITL in the neural retina.

## Disruption of KIT signaling aggravates retinal dysfunction and exacerbates retinal degeneration after LD

The above results prompted us to use the above $Kit^{Wps/Wps}$ mice on a C57BL/6J background to gain insights into the role of KIT signaling specifically during LD of the retina. Even though C57BL/6J mice express the Leu450-Met variant of RPE65 and show reduced susceptibility to LD (*Wenzel et al., 2001*), they have been used as a suitable model of light-induce retinal damage (LIRD) because their susceptibility to LD is similar to that of humans (*Ding et al., 2017*). Thus, we first dark-adapted $Kit^{+/+}$ and $Kit^{Wps/Wps}$ mice and examined them by electroretinography (ERG). Subsequently, the mice were light-exposed (15,000 lux) for 8 days, with pupils dilated by a mydriatic agent once a day, and then re-examined by ERG (*Figure 3A*). As shown in *Figure 3B and C*, under scotopic conditions (rod response), the a- and b-wave amplitudes were similar in $Kit^{+/+}$ and $Kit^{Wps/Wps}$ mice under normal light (426 ± 58µV and 837 ± 1314µV, respectively, in $Kit^{+/+}$, n = 8; and 439 ± 83µV and 863 ± 114µV, respectively, in $Kit^{Wps/Wps}$ mice, n = 8). After LD, the a- and b-wave amplitudes declined in $Kit^{+/+}$ to 294 ± 37µV and 521 ± 27µV, respectively (n = 8) and in $Kit^{Wps/Wps}$ mice to 188 ± 80µV and 365 ± 63µV, respectively (n = 8). These data indicate that the rod-driven circuit was damaged more severely in $Kit^{Wps/Wps}$ retinas compared to $Kit^{+/+}$ retinas. Similarly, under photopic conditions (cone response) (*Figure 3B,D*), the b-wave ERG responses were weakened in $Kit^{Wps/Wps}$ mice more severely (from 89.4 ± 15µV to 40 ± 10µV, n = 8) than those in $Kit^{+/+}$ mice (from 95 ± 13µV to 61 ± 9µV, n = 8), indicating that the cone-driven circuit was also damaged more severely in $Kit^{Wps/Wps}$ retinas. These results suggest that normal KIT signaling helps to protect against retinal damage following LD.

Retinal degeneration analysis revealed that along with the reduction of the electrical responses, LD also led to significant morphological changes, reactive gliosis, Rhodopsin translocation and photoreceptor cell death. As shown in *Figure 3E*, compared to light-damaged $Kit^{+/+}$ retinas, similarly treated $Kit^{Wps/Wps}$ retinas showed a severely reduced thickness of the ONL both in superior and inferior regions (*Figure 3E*, panels a, *F*). In unstressed $Kit^{+/+}$ or mutant retinas, GFAP immunoreactivity, indicative of reactive gliosis, was only detected in the ganglion cell layer, while after light stress, Müller glial cells also became positive for GFAP, but much more so in $Kit^{Wps/Wps}$ compared to $Kit^{+/+}$ mice (*Figure 3E*, panels b). Likewise, Rhodopsin immunoreactivity was not only more severely reduced in light-damaged $Kit^{Wps/Wps}$ compared to $Kit^{+/+}$ retinas but also showed abnormal presence in the ONL (*Figure 3E*, arrows in panels c). These results were consistent with TUNEL assays to evaluate cell death. As shown in *Figure 3E* (panels D) and 3G, although dead cells were present in the ONL of $Kit^{+/+}$ retinas (22 ± 7 per section, n = 10), their number was much higher in $Kit^{Wps/Wps}$ retinas (44 ± 9, n = 10). Hence, these data show that after excessive light exposure, the photoreceptor cells of $Kit^{Wps/Wps}$ mice are not only structurally but also functionally impaired to a greater extent than those of $Kit^{+/+}$ mice. The findings indicate that the $Kit^{Wps}$ mutation accelerates photoreceptor degeneration during LD.

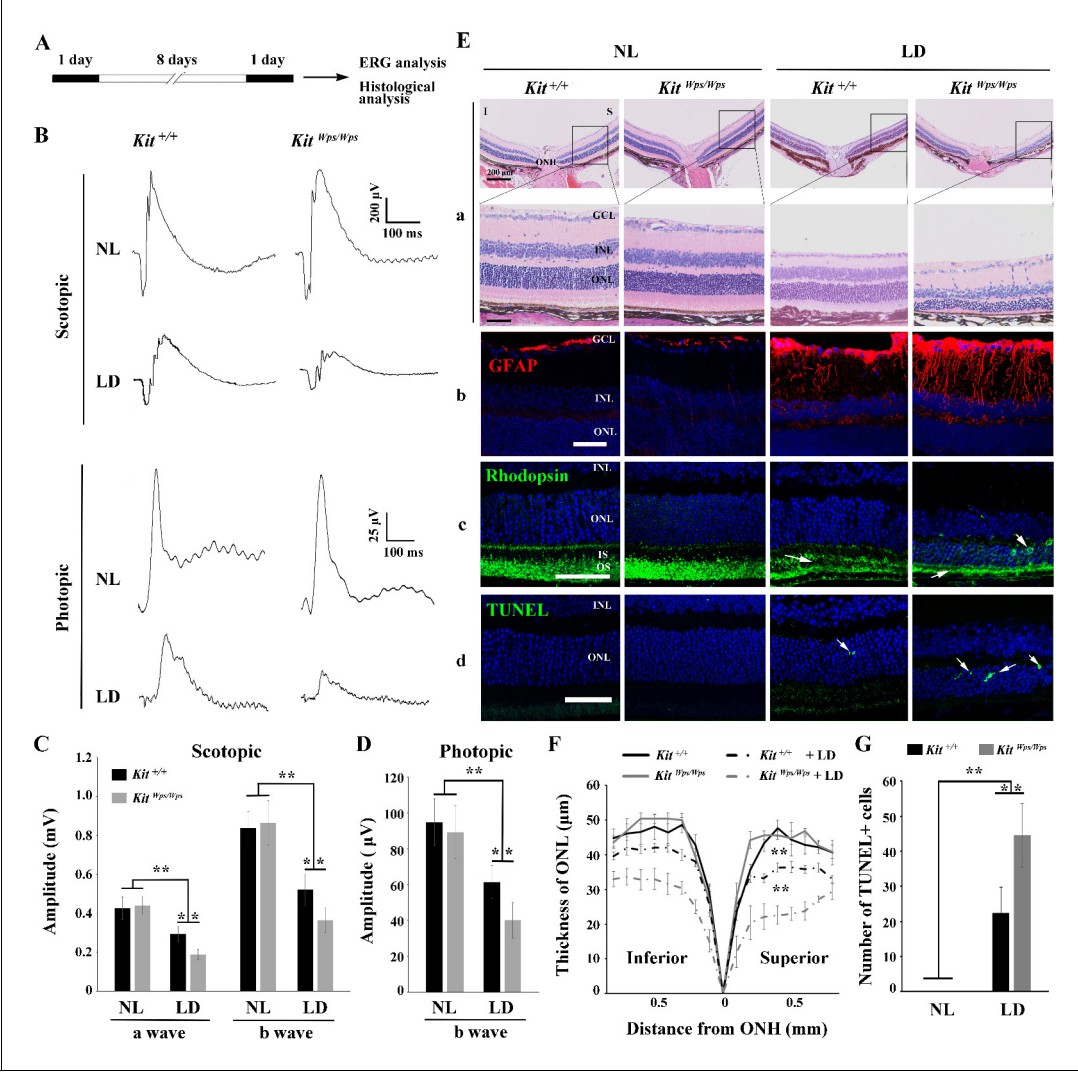

**Figure 3.** *Kit^Wps* mutation significantly damages retinal function and accelerates retinal degeneration during LD. (**A**) Schematic representation of LD treatment of mice for B-G. Three-month-old *Kit^{+/+}* and *Kit^{Wps/Wps}* mice were raised in the dark for one day, and then exposed to constant high intensity light of 15,000 lux for 8 days with mydriasis treatment to induce retinal damage. Retinal functions were subsequently assessed by electroretinography (ERG) after one day dark adaptation. (**B**) ERG traces of *Kit^{+/+}* (left panels) and *Kit^{Wps/Wps}* (right panels) retinas before and after LD treatment under scotopic (upper panels) and photopic conditions (lower panels). Note that both scotopic and photopic responses of the *Kit^{Wps/Wps}* retinas were similar to those of *Kit^{+/+}* retinas with normal light (NL). Usually, the scotopic ERG result is shown as step-wise figure. (**C and D**) Quantification of ERG amplitudes under scotopic (**C**) and photopic conditions (**D**). Note that both scotopic and photopic responses of *Kit^{Wps/Wps}* retinas were impaired more severely than those in *Kit^{+/+}* retinas. Each trace is the average of individual records from at least five mice. (**E**) Retinal degeneration analysis of *Kit^{+/+}* and *Kit^{Wps/Wps}* mice before and after LD treatment by HE staining (**a**), GFAP staining (**b**), Rhodopsin staining (**c**) and TUNEL detection (**d**). Arrows point to the weakened signal or abnormal translocation signal of the rhodopsin (**c**) and dead cells (**d**). (**F**) The curve diagram shows the thickness of ONL from *Kit^{+/+}* and *Kit^{Wps/Wps}* retinas under NL or high-intensity LD conditions. (**G**) Quantification of the number of TUNEL-positive cells in the ONL. GCL, ganglion cell layer; I, inferior; INL, inner nuclear layer; IS, photoreceptor inner segments; ONL, outer nuclear layer; S, superior. ** indicates p<0.01. Scale bar, 50 μm.

The online version of this article includes the following source data for figure 3:

**Source data 1.** Source data for the graphs in Figure 3C, D, G and the diagram in Figure 3F.

## KITL prevents photoreceptor cell death-associated with LD

As the above results indicated a role for KIT signaling in light-induced photoreceptor degeneration, we asked whether KITL may play a protective role in LD. Hence, we overexpressed KITL by intraocular injection of an engineered AAV virus. We first tested in which retinal areas an AAV8-CMV-GFP vector (*Figure 4—figure supplement 1A*) is expressed by scanning fluorescent ophthalmoscopy and

by optical coherence tomography. Fourteen days after virus infection, fundus photographs showed that GFP was found in nearly the entire retina, and cross-sections showed strong GFP signals in photoreceptor cells (*Figure 4—figure supplement 1B*). We then ectopically expressed KITL using an appropriately engineered viral expression vector, AAV8-CMV-KITL (hereafter called AAV8-KITL, schematically shown in *Figure 4A*). Two weeks after infection, we isolated the neural retinas and examined them for KITL expression. Western blots showed a significantly increased expression of KITL (3.3 ± 0.46 fold over uninjected control, n = 4) (*Figure 4B*), and immunostaining showed that overexpression of KITL was almost entirely restricted to photoreceptors in the retinas (*Figure 4C*). These results indicated that KITL can be successfully overexpressed in photoreceptor cells.

In order to test whether overexpression of *Kitl* would affect retinal damage after light exposure, we had to avoid the complication that C57BL/6 mice are homozygous for the $Rpe65^{L450M}$ mutation, which, as mentioned above, leads to increased resistance to LIRD (*Wenzel et al., 2001*). Hence, for these experiments, we used albino mice, which are *Rpe65* wildtype. Without LD, neither AAV8 (schematically shown in *Figure 4—figure supplement 2A*) nor AAV8-KITL caused detectable changes of retinal morphology in these mice (*Figure 4D,E*). After 3 days of LD treatment, retinas infected by AAV8-KITL were thicker than those infected by control AAV8 or left uninfected (*Figure 4D,E*). Furthermore, TUNEL analysis showed that infection with AAV8 control or AAV8-KITL virus did not cause photoreceptor cell death under normal light conditions, but under LD conditions, photoreceptor cell death was less pronounced when AAV8-KITL was used (9 ± 1.6%, n = 6) compared to when control AAV8 was used (40.8 ± 4.5%, n = 6) or the eyes were left uninfected (37.7 ± 5%, n = 6) (*Figure 4F, G*). These results suggest that AAV8-KITL prevents light-induced photoreceptor cell death. Immunostaining indeed showed high level KITL expression in photoreceptor cells infected with AAV8-KITL compared to uninfected retinas (*Figure 4—figure supplement 2B*). Taken together, these data indicate that overexpression of KITL prevents light-induced photoreceptor degeneration in Kit wild type mice.

To investigate whether protection of AAV8-KITL virus indeed depended on KIT, we then used $Kit^{Wps/Wps};Rpe65^{+/+}$ albino mice, obtained by appropriate crossings of $Kit^{Wps}$ heterozygotes with BALB/c albino mice and selection of mice homozygous for $Rpe65^{+/+}$ by PCR. After 3 days of LD treatment, the retinas in both $Kit^{+/+}$ albino and $Kit^{Wps/Wps}$ albino mice showed reduced thickness and near-absence of outer segments, but the effects were more severe in $Kit^{Wps/Wps}$ albino compared with $Kit^{+/+}$ albino mice (*Figure 4H,I*). Importantly, injection of the AAV8-KITL virus prevented light-induced reduction of retinal thickness in $Kit^{+/+}$ albino mice but not in $Kit^{Wps/Wps}$ albino mice (*Figure 4H,I*). Consistent with these results, TUNEL assays showed a cell death rate of 40.6 ± 8.2% in $Kit^{+/+}$ albino mice and 78 ± 11.4% in $Kit^{Wps/Wps}$ albino mice (n = 4) after control infection, while infection with AAV8-KITL significantly reduced the death rate of photoreceptor cells to 8.9 ± 2.1% in $Kit^{+/+}$ albino mice but not in $Kit^{Wps/Wps}$ albino mice, where it remained high (75.9 ± 8.2%, n = 4) (*Figure 4H,J*). Collectively, these data suggest that overexpression of KITL can prevent photoreceptor cell death from LD through activation of the receptor KIT.

## Overexpression of KITL in photoreceptor cells prevents light-induced retinal degeneration

Although AAV-KITL drives overexpression of KITL predominantly in photoreceptor cells, a protective effect from expression in other cells, even if minor, cannot be entirely excluded. Hence, to test whether the protective effect of KITL is indeed cell-autonomous in photoreceptor cells, we expressed KITL selectively in RPE or photoreceptor cells, using either the RPE-specific promoter RPE65 (AAV8-RPE65-KITL-T2A-Flag, hereafter named AAV8-RPE65-KITL, schematically shown in *Figure 5A*) or the photoreceptor cell-specific promoter RHO (AAV8-RHO-KITL, schematically shown in *Figure 5B*). As shown in *Figure 5C and D*, at 2 weeks after injection of AAV8-RPE65-KITL, overexpression of KITL was prominent in the RPE (12.1 ± 2.4 fold, n = 4), but its protective effect against LD was very limited (*Figure 5E,F*). In fact, the death rate of photoreceptor cells was 26.6 ± 5%, n = 8 after infection with AAV8-RPE65-Flag and only slightly reduced to 21.7 ± 2.8%, n = 8 after infection with AAV8-RPE65-KITL (*Figure 5—figure supplement 1A-a and B*). Hence, overexpression of KITL in RPE does not play a major role in prevention of light-induced photoreceptor degeneration.

Infection with AAV8-RHO-KITL led to overexpression specifically in photoreceptor cells as shown by WB (increase over control by 4.23 ± 0.68 fold, n = 4) and IF (*Figure 5G and H*), confirming previous reports (*Allocca et al., 2007*). In contrast to overexpression in RPE, however, overexpression of

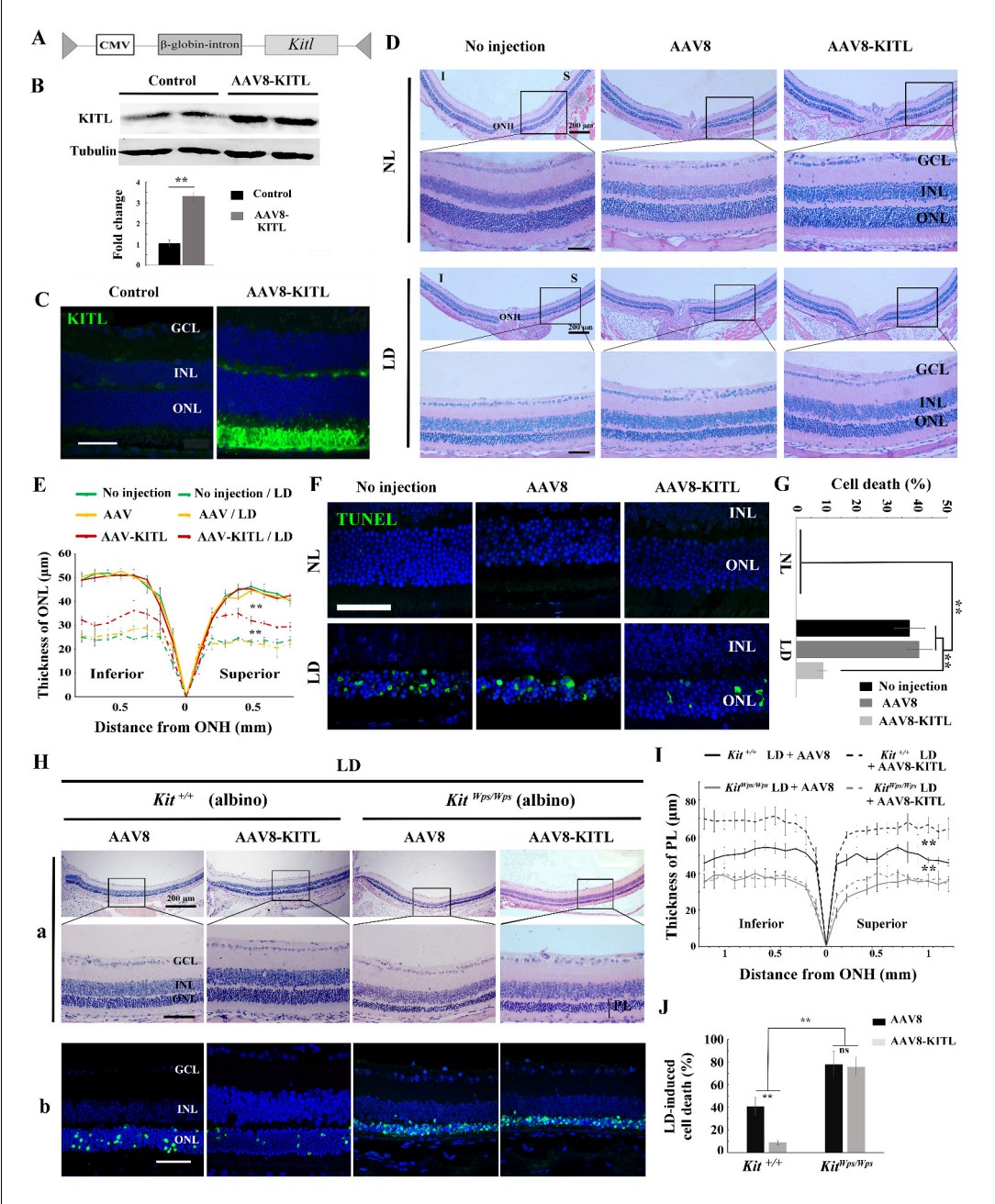

**Figure 4.** Overexpression of KITL by AAV8-KITL virus prevents light-induced retinal degeneration. (**A**) Schematic representation of the AAV8-KITL construct. (**B**) Western blots (upper panels) show the expression of KITL in the retina 2 weeks after intraocular injection of AAV8-KITL. The bar graph (lower panels) shows the relative expression of KITL based on the results of western blots. (**C**) Immunostaining images show the immunoreactivity of KITL in 3-month-old *Kit*[+/+] retinas 2 weeks after intraocular injection of AAV8-KITL. (**D**) Histological images of H and E staining from 3-month-old albino retinas infected with AAV8 or AAV8-KITL virus for 2 weeks and then kept under NL (upper panels) or high intensity LD (15,000 lux) continuously for 3 days (lower panels). (**E**) The curve diagram shows the thickness of ONL from albino retinas under NL or high-intensity LD conditions. (**F, G**) Images of TUNEL assays from the retinas (**F**) and bar graph show the cell death rate of photoreceptor cells (**G**) under the indicated conditions. (**H**) Retinal degeneration analysis of 3-month-old *Kit*[Wps/Wps] albino retinas infected with AAV8 or AAV8-KITL virus for 2 weeks and then kept under high-intensity LD by HE staining (upper panels) and TUNEL detection (down panels). (**I**) The curve diagram shows the thickness of the photoreceptor cell layer from the indicated retinas. (**J**) Quantification of the rate of cell death in the ONL of the indicated retinas. GCL, ganglion cell layer; INL, inner nuclear layer; ONL outer nuclear layer; ONH, optic nerve head; PL, photoreceptor cell layer. ** indicates p<0.01. Scale bar: 50 μm.

The online version of this article includes the following source data and figure supplement(s) for figure 4:

**Source data 1.** Source data for the diagrams in Figure 4E, I and the graphs in Figure 4G, J.

**Figure supplement 1.** AAV8-GFP virus specifically infects photoreceptor cells throughout the *Kit*[Wps] mutant retina.

*Figure 4 continued on next page*

*Figure 4 continued*

**Figure supplement 2.** High expression levels of KITL in LD-treated retina infected with AAV8-KITL virus.

KITL in photoreceptor cells reduced the LD-induced decrease of ONL thickness (*Figure 5I,J*) and prevented photoreceptor cell death (from 25.9 ± 4.7%, n = 8 after infection with AAV8-RHO-Flag to 11.5 ± 2.5%, n = 8 after infection with AAV8-RHO-KITL (*Figure 5—figure supplement 1A-b, C*). Collectively, these results clearly indicate that overexpression of KITL in photoreceptor cells protects these cells more efficiently than overexpression in RPE cells, underscoring the cell-autonomous effect of KITL.

## KIT signaling regulates *Hmox1* expression in photoreceptor cells

Studies of human disease and animal models of photoreceptor degeneration have shown that photoreceptor loss is often accompanied by oxidative damage, including irreversible oxidation of DNA and other biomolecules (*Cuenca et al., 2014*). In addition, Kit signaling has been identified to regulate mitochondrial function and energy expenditure in brown adipose tissue and skeletal muscle, suggesting that Kit signaling may play a role in oxidation (*Huang et al., 2014*). Hence, in order to analyze the mechanism by which LD leads to the demise of photoreceptor cells, we focused on oxidative damage. We first analyzed whether a series of antioxidant and wound healing genes would show expression changes in $Kit^{+/+}$ retinas following LD (*Figure 6A*). In fact, after LD, the expression of 17 such genes, including *Axl* and *Hmox1*, was changed in $Kit^{+/+}$ retinas (*Figure 6B*). The protein product of *Hmox1*, HMOX1, catalyzes the degradation of heme into three biologically active end products, biliverdin/bilirubin, CO and ferrous ion (*Otterbein et al., 2016*). It is known to be induced by oxidative stress in multiple tissues as well as by LD of the retina (*Kutty et al., 1995*) and thought to act as an antioxidant (*Otterbein et al., 2016*). Under normal light/dark cycle conditions, HMOX1 levels analyzed on western blots (*Figure 6C*) or by immunostaining (*Figure 6D*) were similarly low in both $Kit^{+/+}$ and $Kit^{Wps/Wps}$ retinas. After LD, however, HMOX1 expression was significantly upregulated in $Kit^{+/+}$ retinas (3.6 ± 0.6 fold) but only slightly in $Kit^{Wps/Wps}$ retinas (1.4 ± 0.5 fold, n = 3) (*Figure 6C,D*). These data indicate that LD-dependent upregulation of *Hmox1* is markedly influenced by KIT signaling.

The importance of KIT signaling for the regulation of HMOX1 expression can also be demonstrated in vitro in 661 W cells, which are derived from a medulloblastoma and whose gene expression profiles suggests a photoreceptor origin (*Tan et al., 2004*). In fact, 661 W cells have been widely used as a tool for investigating photoreceptor cell biology and function (*Tan et al., 2004*; *Wheway et al., 2019*). To test for the differential effects of Kit-WT and KIT-WPS, we transfected such cells with corresponding expression vectors and treated the cells with KITL. As expected, the ACK2 antibody, which reacts with the extracellular portion of KIT, readily detects KIT after transfection of either of the two KIT vectors (*Figure 6—figure supplement 1A*). Western blots indicated an increase of 2.9 ± 0.23 fold with Kit-WT and 2.2 ± 0.9 fold with Kit-Wps (*Figure 6—figure supplement 1B*). Transfection with KIT also led to upregulation of HMOX1, although with a clear difference between KIT-WT (6.2 ± 0.86 fold, n = 4) and KIT-Wps (3 ± 0.6 fold, n = 4). These results are consistent with the in vivo results described above.

Given the in vitro results, we then asked whether ectopic activation of KIT signaling would induce *Hmox1* expression in the retina without LD. Hence, we analyzed the expression of HMOX1 in retinas of $Kit^{+/+}$ mice kept under normal light conditions and infected with AAV8-KITL virus. One week after injection, western blots showed a correlation between high levels of KITL (4.6 ± 0.9 fold induction, n = 4) and HMOX1 (2.9 ± 0.5 fold induction, n = 4) (*Figure 6E*). Consistent with this observation, immunostaining showed that the expression of KITL was dramatically enhanced in the superior retinal region around the location of injection, coinciding with high level HMOX1 expression (*Figure 6F*). These data suggest that activation of KIT signaling can induce HMOX1 expression in $Kit^{+/+}$ retinas independent of LD treatment.

## KITL/KIT signaling activates NRF2 to regulate *Hmox1* expression

To understand how KIT regulates *Hmox1* expression, we analyzed whether KIT signaling is able to regulate the expression of the transcription factor NRF2, an upstream regulator of *Hmox1* that is

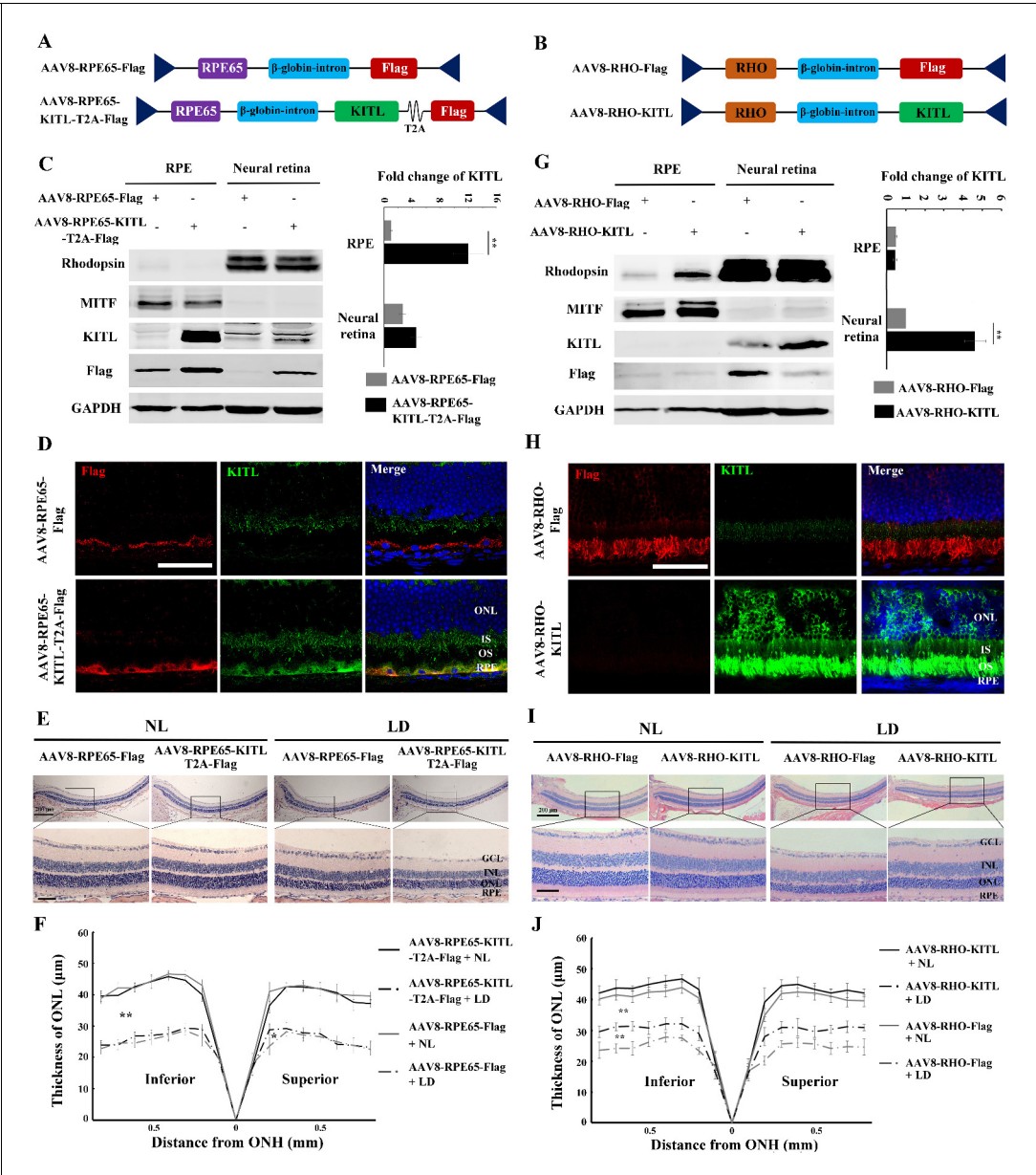

**Figure 5.** Overexpression of KITL by AAV8-RHO-KITL but not AAV8-RPE65-KITL virus prevents light-induced retinal degeneration. (A) Schematic representation of the AAV8-RPE65-Flag and AAV8-RPE65-KITL-T2A-Flag constructs. (B) Schematic representation of the AAV8-RHO-Flag and AAV8-RHO-KITL constructs. (C) Western blots showing expression of KITL in the RPE and neural retina 2 weeks after intraocular injection of AAV8-RPE65-Flag or AAV8-RPE65-KITL-T2A-Flag (left panels). The bar graph shows the relative expression levels of KITL based on the results of western blots (right panels). (D) Immunoreactivity of KITL in retina sections of 2-month-old albino mice 2 weeks after intraocular injection of AAV8-RPE65-KITL-T2A-Flag or control AAV8-RPE65-Flag. (E) Histological images of H and E staining of retinas infected with the indicated viruses for 2 weeks and then kept under NL or high-intensity LD (15,000 lux) continuously for 3 days. (F) The curve diagram shows the thickness of the ONL from albino retinas under the indicated condition. (G) Western blots (left panels) and the bar graph (right panels) show the expression of KITL in the RPE and neural retina at 4 weeks after intraocular injection of AAV8-RHO-Flag or AAV8-RHO-KITL virus. (H) Immunoreactivity of KITL in retina sections of 2-month-old albino mice 4 weeks after intraocular injection of the AAV8-RHO-Flag or AAV8-RHO-KITL. (I, J) Histological images of H and E staining (I) and the curve diagram (J) showing retinal degeneration under the indicated conditions. GCL, ganglion cell layer; INL, inner nuclear layer; ONL outer nuclear layer; IS, inner segment; OS, outer segment; ONH, optic nerve head. * or ** indicates p<0.05 or p<0.01. Scale bar: 50 μm.

The online version of this article includes the following source data and figure supplement(s) for figure 5:

**Source data 1.** Source data for the diagrams in Figure 5E, J.

**Figure supplement 1.** Overexpression of KITL in photoreceptor cells prevents light-induced photoreceptor cell death.

**Figure supplement 1—source data 1.** Source data for the graphs in Figure 5-figure supplement 1B, C.

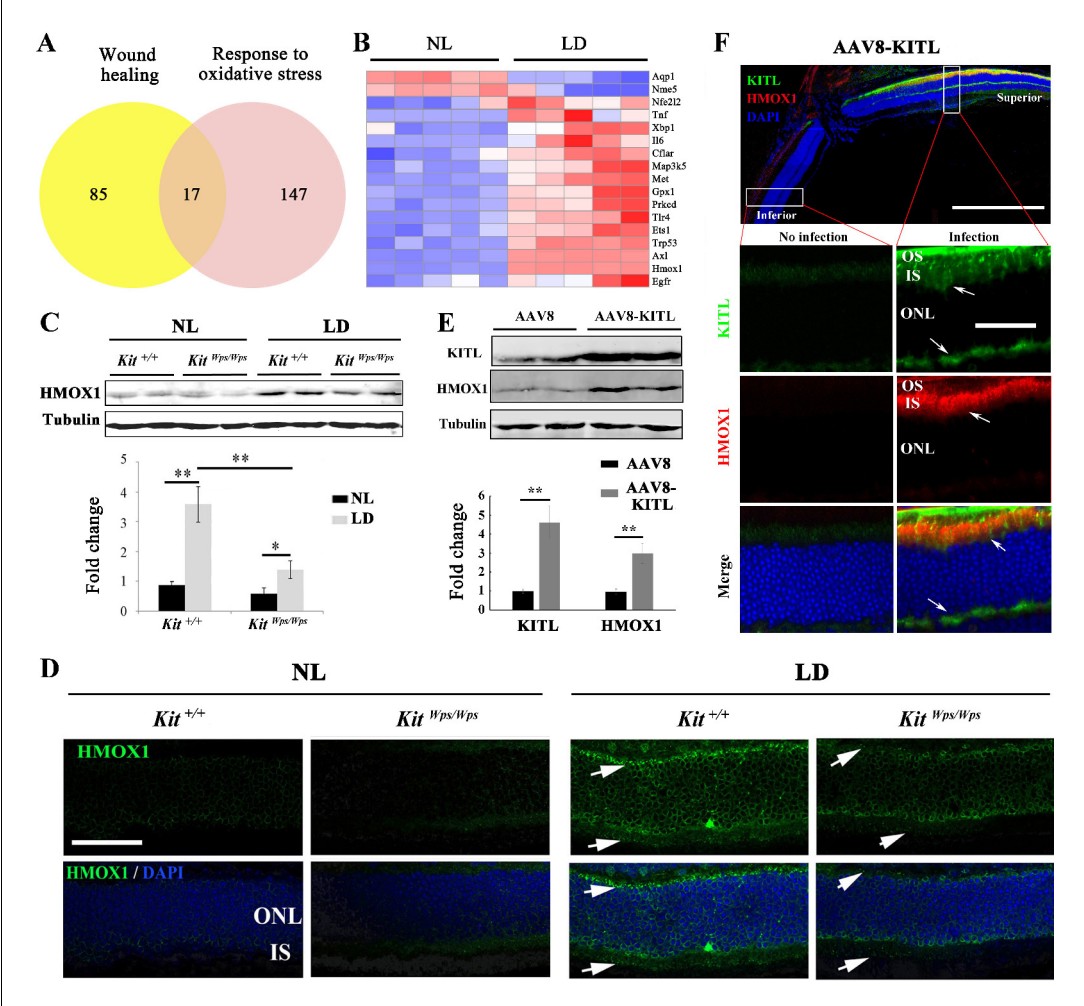

**Figure 6.** KIT signaling regulates expression of *Hmox1* in photoreceptor cells. (**A**) Venn diagram showing numbers of differentially expressed genes involved in response to oxidative stress and/or wound healing. Results from *Kit+/+* retinas after LD for 1 day. (**B**) Heat-map of differentially expressed genes from the 17 genes common between the wound healing group and response to oxidative stress group according to the result of Venn diagram. Messenger RNA expression changes of anti-oxidant genes in *Kit*[+/+] retinas after LD for 1 day. (**C and D**) Expression of HMOX1 in *Kit*[+/+] and *Kit*[Wps/Wps] retinas were detected by western blots and immunostaining. Note that the expression of HMOX1 was more significantly upregulated in light-degenerated *Kit*[+/+] compared to light-degenerated *Kit*[Wps/Wps] photoreceptors. (**E**) Western blots showing the expression of KITL and HMOX1 in the retinas infected with control AAV8 or AAV8-KITL virus. Bar graphs show the relative expression of KITL and HMOX1 in retinas based on the results of western blots. (**F**) Immunoreactivity of KITL and HMOX1 in *Kit*[+/+] retina sections 1 week after subretinal injection of AAV8-KITL. IS, photoreceptor inner segments; ONL, outer nuclear layer; OS, photoreceptor outer segment. * or ** indicates p<0.05 or p<0.01. Scale bar: 50 μm.

The online version of this article includes the following figure supplement(s) for figure 6:

**Figure supplement 1.** Overexpression of KIT upregulates *Hmox1* expression in 661W photoreceptor cells.

activated in many cell types in response to stressful environments (*Sies et al., 2017*; *Kleszczyński et al., 2016*). Although western blots showed no change in NRF2 expression in retinas after light treatment (*Figure 7A,B*), immunostaining indicated a dramatic increase in nuclear accumulation of NRF2 in photoreceptor cells after light treatment for 8 days (*Figure 7C*). This nuclear accumulation was much reduced in *Kit*[Wps/Wps] mice (14 ± 5 per section, n = 6) compared to *Kit*[+/+] mice (28 ± 9, n = 6) (*Figure 7C,D*), suggesting that Kit signaling regulates nuclear translocation of NRF2 in photoreceptor cells.

To further examine whether NRF2 nuclear accumulation was dependent on KITL, we used *Kitl*-specific siRNAs to reduce the expression of *Kitl* in 661W photoreceptor cells (*Figure 7—figure supplement 1*). Indeed, immunostaining revealed that both Kitl-si-RNAs increased the number of cells with cytosolic NRF2 expression from 4.74 ± 2.27% to 30.2 ± 8.3% or 36.8 ± 2.24% (n = 3),

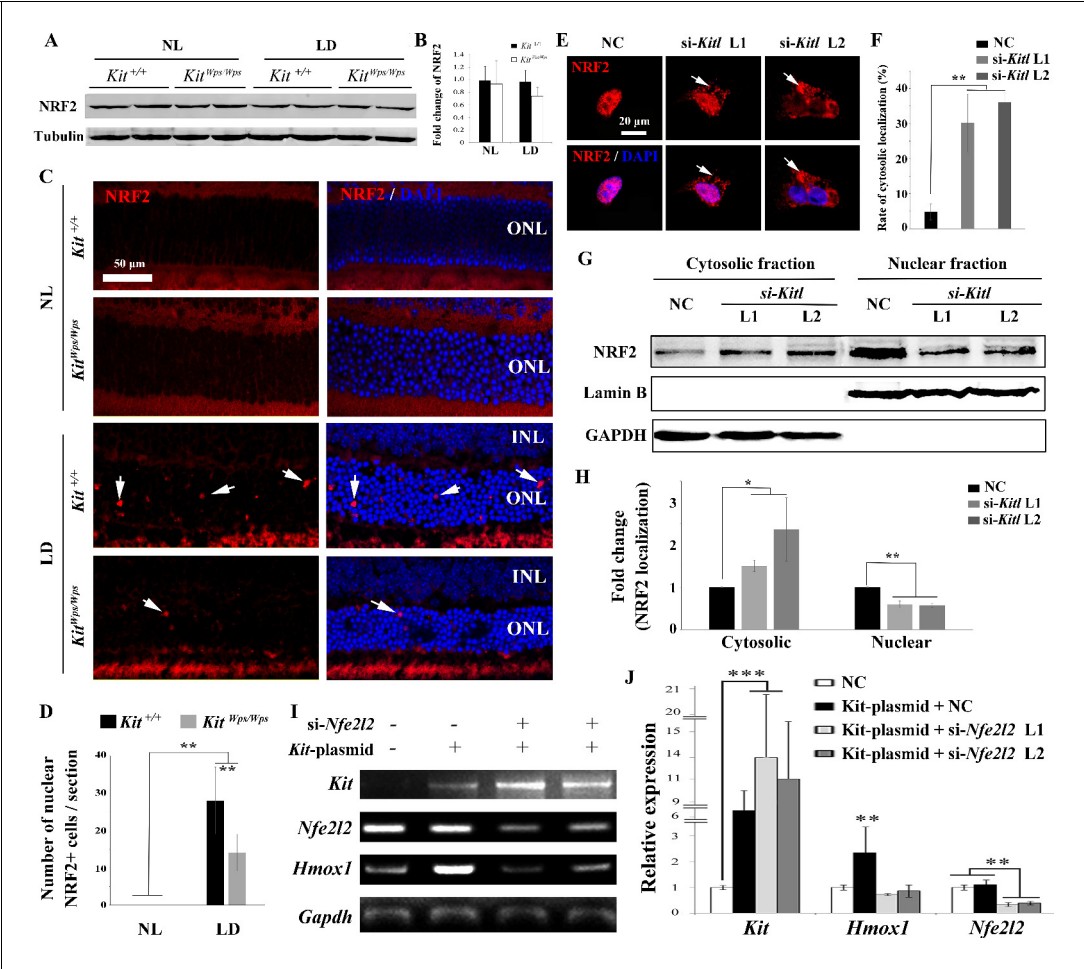

**Figure 7.** KIT signaling acts through the transcription factor NRF2 to regulate *Hmox1* expression. (**A**) Western blots for analyzing NRF2 expression in *Kit*[+/+] and *Kit*[Wps/Wps] retinas under the indicated conditions. (**B**) The bar graph shows quantification of NRF2 expression in the indicated retinas. (**C**) Immunostaining of anti-NRF2 in *Kit*[+/+] and *Kit*[Wps/Wps] retinas under the indicated conditions. The arrows point to nuclear signals of NRF2 in the ONL. (**D**) Quantification of the number of nuclear NRF2 positive cells in the retina. Note that LD induced nuclear accumulation of NRF2 in photoreceptor cells. (**E–H**) Analysis of subcellular localization of NRF2 in 661W photoreceptor cells treated with si-*Kitl* by immunostaining (**E**, and **F**) and western blot (**G**, and **H**). Note that knockdown of *Kitl* led to an increase in the proportion of cells with cytosolic NRF2. (**I, J**) Analyses of the regulation of HMOX1 in 661 W cells after overexpression of KIT together with si-*Nfe2l2*. The images of RT-PCR (**I**) and qPCR (**J**) show the expression levels of *Kit*, si-*Nfe2l2*, and *Hmox1* under the indicated treatments. Note that upregulation of *Hmox1* induced by overexpression of KIT was blocked by the knockdown of *Nfe2l2*. * indicates p<0.05, ** indicates p<0.01.

The online version of this article includes the following figure supplement(s) for figure 7:

**Figure supplement 1.** Kitl mRNA expression levels are reduced after transfection with Kitl-si-RNAs in 661 W cells.

respectively (*Figure 7E,F*). In addition, western blots showed that knock down of *Kitl* increased the levels of NRF2 in the cytoplasm (1.5 ± 0.13 fold, n = 3) and decreased the levels of nuclear NRF2 (0.59 ± 0.09 fold, n = 3) (*Figure 7G,H*). Taken together, these data clearly show that KIT signaling regulates the cytoplasmic/nuclear distribution of NRF2 in photoreceptor cells.

To investigate whether the regulation of *Hmox1* by KIT signaling depends on NRF2, we then used siRNA to knock down *Nfe2l2* mRNA that encodes NRF2 protein in 661 W cells overexpressing KIT (*Figure 7I,J*). Consistently, overexpression of KIT did not significantly change *Nfe2l2* expression (1.1 ± 0.2 fold, n = 3) but upregulated *Hmox1* expression (2.35 ± 1 fold, n = 3). This upregulation was blocked by knockdown of *Nfe2l2*. Collectively, these data suggest that KIT signaling activates NRF2 to regulate the expression of *Hmox1* in photoreceptor cells.

## *Hmox1* overexpression partially rescues retinal degeneration in light-damaged *Kit* mutant mice

While the above results demonstrated that KIT signaling regulates *Hmox1* expression in the retina and plays a neuroprotective role during LD, the relevant function of HMOX1 in retina in vivo was still unclear. To analyze whether KIT signaling exerts its protective effect against LD through HMOX1, we ectopically expressed *Hmox1* in *Kit^{Wps/Wps}* albino mice, using an AAV8-based vector engineered to express HMOX1 (*Figure 8A*). Western blotting and immunostaining showed photoreceptor-restricted expression after 2 weeks after virus infection (*Figure 8A,B*), with levels similar to those observed in uninjected *Kit^{+/+}* retinas after LD. ERG measurements showed that without LD, control AAV8 or AAV8-HMOX1 virus did not cause significant changes in the ERG scotopic trace (*Figure 8C,D*). Following 3 days of LD, however, the amplitudes of ERG scotopic traces were depressed after infection with AAV8 or AAV8-HMOX1 virus, but the b-wave amplitudes were higher when AAV8-HMOX1 was used (381 ± 40μV, n = 5) compared to when control AAV8 was used (152 ± 33 μV, n = 5) (*Figure 8C,D*). These results suggest that overexpression of HMOX1 partially rescues retinal function in the *Kit^{Wps/Wps}* albino mice during LD. Furthermore, histological analyses showed that after LD, AAV8-HMOX1-infected retinas were thicker than control AAV8-infected ones, especially with respect to the photoreceptor cell layer (*Figure 8E,F*). These data suggest that over-expression of HMOX1 protects photoreceptor cells in *Kit^{Wps/Wps}* albino retinas against LD and that LD-induced HMOX1 plays an important antioxidant role in retinas in vivo.

To understand whether protection of KITL depends on HMOX1, we used si-*Hmox1* to knockdown the expression of *Hmox1* in 661W photoreceptor cells under light stress. Western blots showed that both Hmox1-si-RNAs downregulate *Hmox1* expression by 0.5 ± 0.2 fold and 0.46 ± 0.23 fold (n = 3), respectively (*Figure 8—figure supplement 1A,B*). We then tested whether knockdown of *Hmox1* would affect in vitro light-induced cell death. As shown in *Figure 8—figure supplement 1C and D*, cell numbers were similar in each group before LD but dramatically decreased in the control group (from 409 ± 44 to 147 ± 15 (n = 3) after 8 hr of LD treatment). Addition of exogenous KITL (5.6 nM) was able to prevent cell loss caused by LD to 278 ± 28 (n = 3). This preventive effect was totally blocked by knockdown of *Hmox1* via Hmox1-si-RNAs (cell loss 147 ± 16, n = 3 and 131 ± 17, n = 3, respectively). In addition, the data of propidium iodide (PI) staining also showed that stimulation of KITL decreased light-induced cell death rate from 83 ± 4.6% to 40 ± 5.3% (n = 3), but this protection by KITL disappeared when the expression of *Hmox1* was knocked down by Hmox1-si-RNAs (*Figure 8—figure supplement 1E*). These results suggest that protection of KITL against light-induced photoreceptor cell loss depends on HMOX1.

## KITL preserves photoreceptors and restores retinal function in genetic mouse models of retinal degeneration

We finally evaluated whether KITL might provide neuroprotection in mouse models of human retinitis pigmentosa (RP). Mice homozygous for a mutation in the gene encoding the photoreceptor cGMP phosphodiesterase, *Pde6b^{rd10}* (hereafter called *rd10/rd10*) share many characteristics of human retinal degeneration resulting from a mutation in the homologous gene (*Wang et al., 2018*). Due to the fact that retinas of *rd10/rd10* mice degenerate soon after birth, we performed subretinal injection of AAV8-KITL virus into the eyes of such mice (n = 5) at postnatal day 3 and examined their retinal functions at postnatal day 24. As shown in *Figure 9A*, ectopic expression of KITL was observed at 3 weeks in *rd10/rd10* photoreceptor cells in the eye infected with AAV8-KITL, but not in those of the control-injected contralateral eye. These results suggest that subretinal injection of AAV8-KITL virus at P3 can mediate overexpression of KITL in photoreceptor cells of *rd10/rd10* mice. We then analyze the effects of KITL overexpression on retinal degeneration. The thickness of the ONL was significantly increased from the peripheral region to the central region after infection with the AAV8-KITL virus (*Figure 9B,C*). In P24 control-injected *rd10/rd10* retina, Rhodopsin was abnormally translocated from the OS to the ONL, a phenomenon that was partially rescued by infection with the AAV8-KITL virus (*Figure 9D*). TUNEL analysis showed a large number of dead cells in the ONL of *rd10/rd10* mice infected with control AAV8 virus (death rate, 16.2 ± 2%, n = 5), but a significantly reduced number after AAV8-KITL virus infection (8.9 ± 1.6%, n = 5) (*Figure 9E*). We next examined the effects of KITL on retinal function of *rd10/rd10* mice. The ERG analysis revealed that the scotopic traces of control eyes presented nearly horizontal lines (b-wave amplitude,

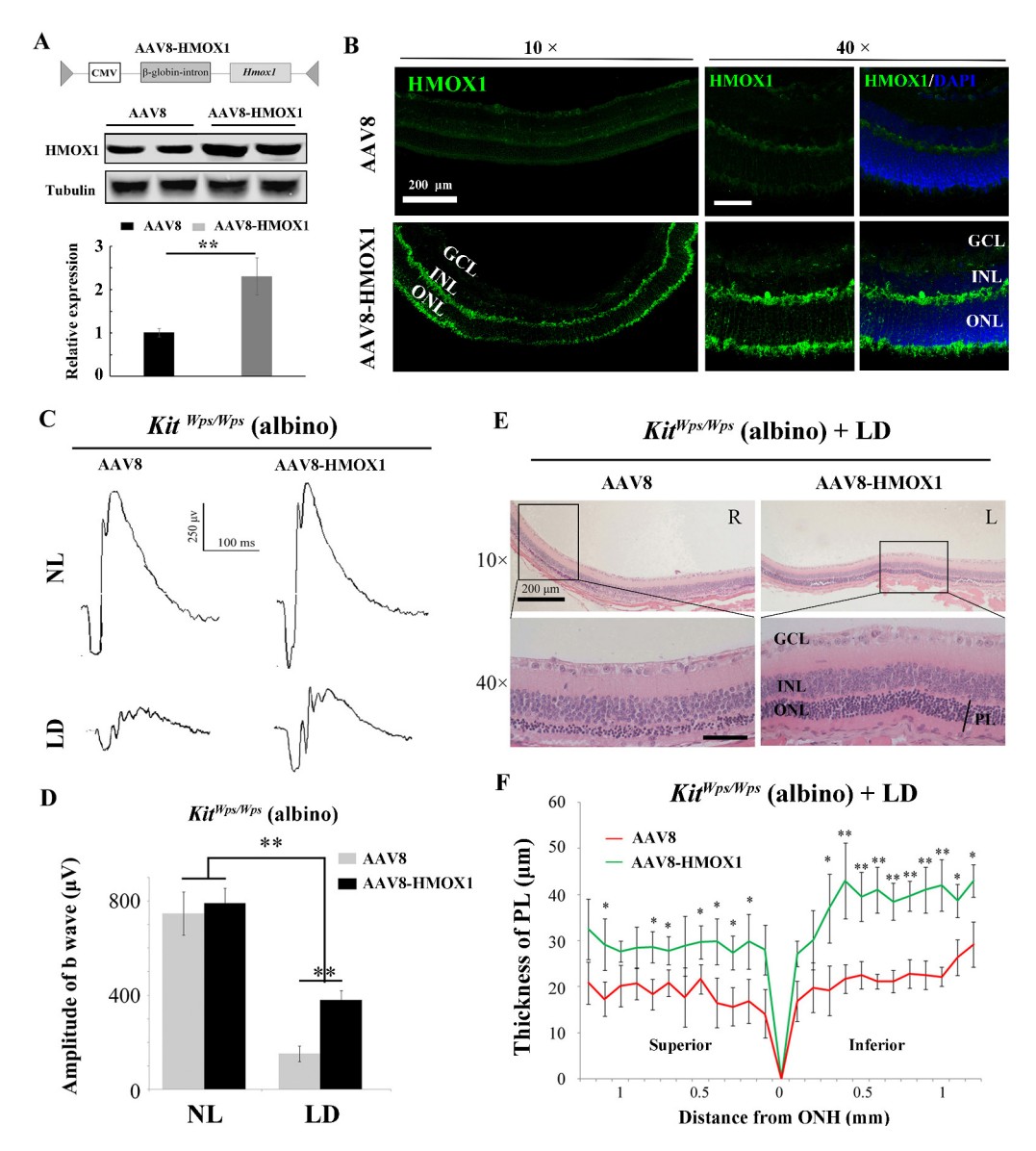

**Figure 8.** Protective role of HMOX1 in LD-treated *Kit*^{Wps/Wps} retina. (**A**) Schematic representation of the AAV8-HMOX1 construct (upper panel) and western blots show the expression of KITL in the retina 2 weeks after subretinal injection of AAV8- HMOX1 virus (lower panels). The bar graph shows the relative expression of HMOX1 based on the results of western blots. (**B**) Ectopic HMOX1 expression in retina by infection of AAV8 or AAV8-HMOX1 virus was visualized by immunofluorescence (n = 4). (**C**) ERG scotopic traces of *Kit*^{Wps/Wps}; albino mice infected with control AAV8 (n = 5) or AAV8-HMOX1 (n = 5) virus for 2 weeks and then kept under NL or high-intensity LD (15,000 lux) for 3 days. (**D**) Quantification of amplitude of b-wave from standard response based on the results from C. (**E**) Histological analysis of *Kit*^{Wps/Wps}; albino retina infected with control AAV8 (n = 6) or AAV8-HMOX1 (n = 6) virus after LD. (**F**) Curve diagram showing the total thickness of the ONL under the indicated conditions. GCL, ganglion cell layer; INL, inner nuclear layer; ONL, outer nuclear layer; PL, photoreceptor cell layer. * or ** indicates p<0.05 or p<0.01. Scale bar: 50 μm.

The online version of this article includes the following source data and figure supplement(s) for figure 8:

**Source data 1.** Source data for the graphs in Figure 8D and the diagram in Figure 8F.

**Figure supplement 1.** Protective role of KITL depends on HMOX1 in light-induced photoreceptor cell death.

142.2 ± 43.60.4μV, n = 5), while infection with the AAV8-KITLvirus led to clear a- and b-waves (b-wave amplitude, 304 ± 30μV, n = 5) (*Figure 9F,G*), suggesting that overexpression of KITL partially restores the retinal function in *rd10/rd10* mice.

In order to further confirm the protective role of KITL in hereditary photoreceptor degeneration, we used an additional RP model, mice harboring a different mutant allele of *Pde6b*, *Pde6b*^{rd1}

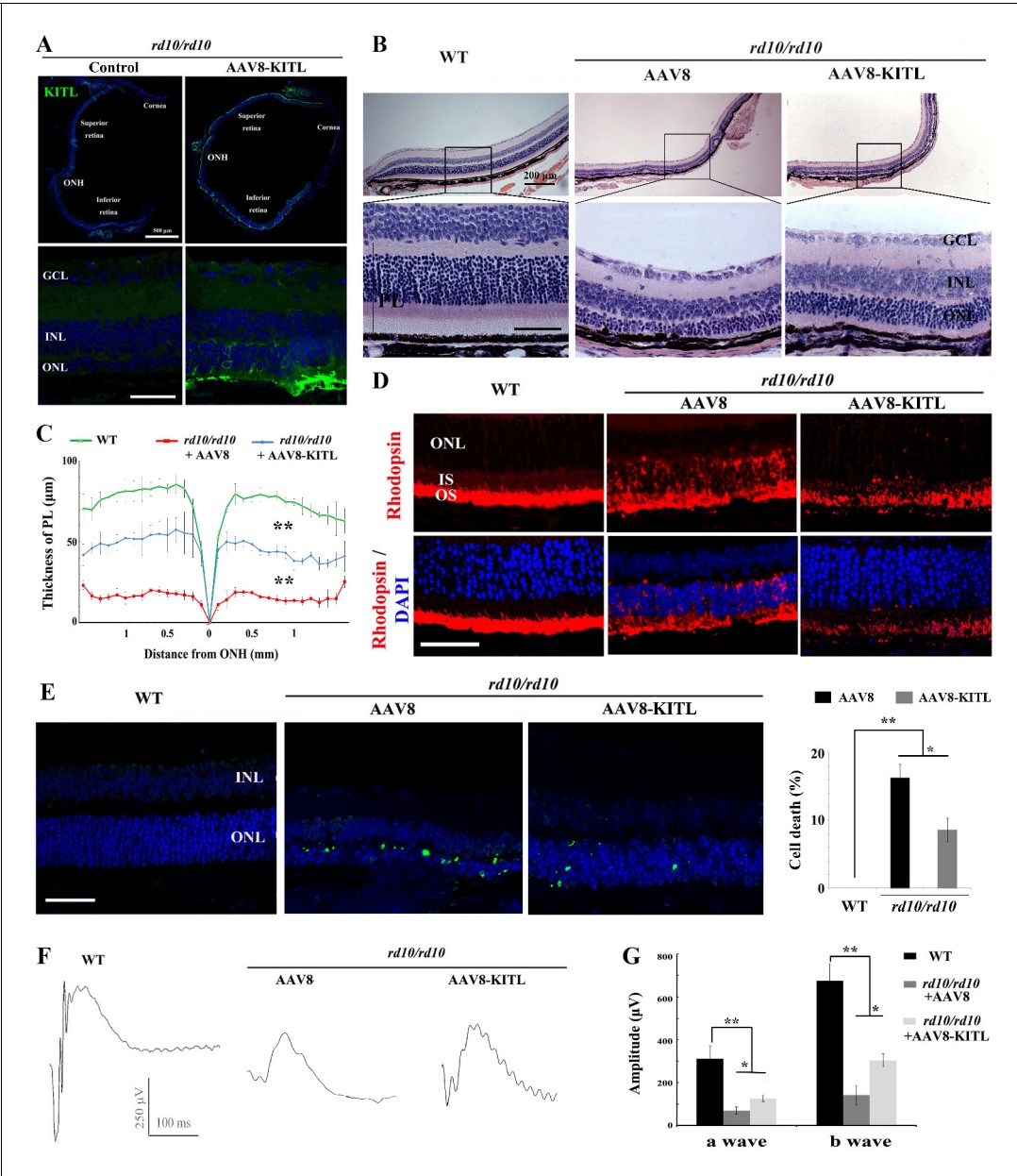

**Figure 9.** Ectopic expression of KITL prevents retinal degeneration of *rd10* homozygous mice. (**A**) Expression of KITL in *rd10/rd10* retinas 2 weeks after infection with AAV8-KITL virus or control was visualized by immunofluorescence. Note that the subretinal injection was performed in *rd10/rd10* mice at postnatal day 3. (**B and C**) Histological analysis of *rd10/rd10* retinas infected with AAV8 or AAV8-KITL virus (n = 5) or control (**B**). Curve diagram showing the thickness of total photoreceptor cell layer from *rd10/rd10* retina under the indicated conditions (**C**). (**D**) Rhodopsin expression in WT and *rd10/rd10* retina under the indicated conditions. (**E**) Apoptosis of photoreceptor cells from *rd10/rd10* mice with of AAV8 or AAV8-KITL was analyzed by TUNEL assay and is presented as fluorescent images (upper panels) and as bar graphs after quantitation (lower panels). (**F, G**) ERG scotopic traces (**F**) obtained from *rd10/rd10* mice with infection of AAV8-KITL virus (n = 5) or control (n = 5) and kept under normal conditions for 25 days (right panels). Bar graph (**G**) shows the quantification of the amplitude of the b-wave from standard response based on the results of the ERG scotopic traces (**F**). GCL, ganglion cell layer; INL, inner nuclear layer; ONL, outer nuclear layer; PL, photoreceptor layer. * or ** indicates p<0.05 or p<0.01. Scale bar: 50 μm.

The online version of this article includes the following source data and figure supplement(s) for figure 9:

**Source data 1.** Source data for the diagram in Figure 9C and the graphs in Figure 9E, G.

**Figure supplement 1.** Ectopic expression of KITL prevents retinal degeneration of *rd1* homozygous mice.

**Figure supplement 1—source data 1.** Source data for the graphs in Figure 9-figure supplement 1B and the diagrams in Figure 9-figure supplement 1D, E.

(hereafter called *rd1/rd1*) that was kept on a different background, FVB. The results were similar to those obtained with *rd10/rd10* mice (**Figure 9—figure supplement 1**). Taken together, these results strongly indicate that overexpression of KITL prevents photoreceptor cell death and partially restores retinal function not only in LD but also genetic models of retinal degeneration.

## Discussion

Several lines of evidence support the notion that activation of KIT signaling in photoreceptor cells activates NRF2, in turn inducing expression of HMOX1, and protects these cells against light-induced degeneration. First, KIT is expressed in mouse retina, including in its photoreceptor cells, and its activation leads to stimulation of the MAPK and the PI3K/AKT pathways. Second, photoreceptor cells respond to prolonged high-level light exposure by upregulating their endogenous KITL expression, which activates KIT signaling in these cells. Third, when KITL is experimentally overexpressed in *Kit*$^{+/+}$ retina, it alleviates the effects of LD. Fourth, when KIT signaling is genetically impaired, photoreceptor cells are more susceptible to LD, and this enhanced susceptibility cannot be overcome by additional experimental upregulation of KITL. Fifth, experimental expression of HMOX1 in photoreceptor cells in vivo partially protects these cells against light damage even when their KIT signaling pathway is impaired. Sixth, experimental overexpression of KITL also partially protects against photoreceptor cell loss in *Kit*$^{+/+}$ mice harboring mutations in *Pde6b*, the mouse homolog of the human RP gene, whose mutations are associated with retinal degeneration in both mice and humans.

Given these findings, the use of KITL and activation of KIT signaling may add to the arsenal of endogenous pathways that may be considered clinically for photoreceptor protection during retinal degeneration. Indeed, in contrast to other strategies such as stem cell therapies, optogenetics and retinal prostheses, which all face major technical and financial investments (**Scholl et al., 2016**), the application of endogenous neuroprotective factors would seem to provide a more straightforward therapeutic avenue, with the added benefit that it may work regardless of the actual pathogenetic mechanism underlying photoreceptor degradation (**Pardue and Allen, 2018**). AAV-mediated expression of neurotrophic factors or antioxidant genes such as RdCVF and NRF2 has indeed proved beneficial in protecting photoreceptors (**Leveillard and Sahel, 2010**; **Liang et al., 2017 Xiong et al., 2015**), and similar strategies using KIT activation may prove equally valuable.

Previous studies have indicated, however, that Kit signaling can have both beneficial and deleterious effects (**Lennartsson and Rönnstrand, 2012**). On the one hand, KIT is needed for cell development and survival (**Blume-Jensen et al., 2000**; **Hou et al., 2000**; **Wen et al., 2010**), but on the other hand, excessive KIT signaling, associated, for instance, with activating *KIT* mutations, may also lead to formation of tumors, in particular gastrointestinal stromal tumors, adult acute myeloid leukemia, and melanoma (**Lennartsson and Rönnstrand, 2012**). Hence, activation of KIT signaling should be used with caution in clinical settings. Nevertheless, local expression of KITL following intraocular injection of recombinant AAV might considerably reduce potential deleterious effects because gene expression is locally restricted and transfer of protein to other tissues is effectively blocked by the blood retinal barrier. In addition, the optimal choice of virus promoters and doses reduces the risk of virus-induced retinal damage (**Xiong et al., 2019**). Nevertheless, before any further use of KITL in a clinical setting is entertained, it is paramount to establish the precise pathway of its action in the retina.

In the current paper we have so far shown that it is KITL activation in photoreceptor cells themselves, rather than in RPE cells or potentially other retinal cells, that has protective effects on retinal degeneration, and that it is cell-autonomous KITL action that protects photoreceptor cells against degeneration. Nevertheless, other cell types also express KIT (**Koso et al., 2007**; **Zou et al., 2019**; **Too et al., 2017**). Among them are Müller glial cells, which have been reported to participate in reactive gliosis and regulate retinal repair during retinal injury (**Too et al., 2017**; **Goldman, 2014**). Therefore, it cannot be excluded that cells such as Müller glial cells indirectly help to mediate KITL action on photoreceptor cells. Such an indirect, non-cell autonomous effect of a cellular receptor, EDNRB (the receptor for endothelin-3), has been observed, for instance, in neural crest-derived melanocytes (**Hou et al., 2004**).

We have also shown that the pathway downstream of KIT involves a cytoplasm-to-nuclear shift of the transcription factor NRF2, which is likely mediated by KIT signaling-dependent activation of AKT

(*Yoo et al., 2017*). Moreover, KIT signaling leads to increased expression of HMOX1 in an NRF2-dependent way. The involvement of HMOX1 seems particularly interesting as it can reduce oxidative stress by breaking down heme and reducing molecular oxygen and oxygen radicals in the heme oxygenase reaction pathway (*Otterbein et al., 2016*). Photoreceptor cells have a rich complement of mitochondria to accommodate their high metabolic rate and are more sensitive to oxidative stress than many other cell types (*Campochiaro and Mir, 2018*). The development of AMD, for instance, is tightly linked to oxidative damage (*Mitchell et al., 2018*), and antioxidant therapy can delay cell loss in retinal degenerative diseases and lead to reduced photoreceptor cell death in experimental animals (*Donovan et al., 2001*; *Haruta et al., 2009*). Oxidative stress has also been observed in many other neurodegenerative conditions, including LIRD, and antioxidant treatment is useful in their prevention (*Donovan et al., 2001*; *Suzuki et al., 2012*; *Xiong et al., 2015*). The induction of *Hmox1* following LD (*Hadziahmetovic et al., 2012*) can therefore be seen as a homeostatic mechanism operating in photoreceptor cells (*Sun et al., 2007*). Nevertheless, it is important to recognize that in our model, KITL overexpression or isolated overexpression of HMOX1 did not lead to full protection against LD. Hence, the KITL-KIT-NRF2-HMOX1 axis that we here identified is likely only one among many pathways involved in protection of photoreceptor cells against LD.

In summary, our findings provide strong evidence for KIT signaling in protecting photoreceptor cells against light-induced retinal damage as well as against inherited forms of retinal degeneration. We believe that these findings justify a detailed exploration of how this signaling pathway could be harnessed pharmacologically for prevention or delay of progressive photoreceptor cell and vision loss as seen in AMD, RP and other retinal degenerations. Hence, the findings have implications not only for our understanding of the relationship between genetic and environmental factors in retinal degeneration in general but also as a potential treatment for photoreceptor loss in humans.

## Materials and methods

### Mice and histology

All animal experiments were carried out in accordance with approved guidelines of the Wenzhou Medical University Institutional Animal Care and Use Committee (Permit Number: WZMCOPT-090316). C57BL/6J mice were obtained from The Jackson Laboratory and $Kit^{Wps}$ mice carrying a point mutation in the extracellular domain of KIT on a C57BL/6J background were as previously described (*Guo et al., 2010*). Because $Kit^{Wps}$ homozygotes are infertile, crosses between $Kit^{Wps}$ heterozygotes were set up to obtain $Kit^{Wps/Wps}$ mice. To obtain appropriate $Kit^{Wps/Wps}$ albino mice, we crossed B6-$Kit^{Wps/+}$;$Rpe65^{L450M/L450M}$ mice with BALB/c albino mice ($Tyr^c/Tyr^c$;$Rpe65^{+/+}$) to eventually obtain $Kit^{Wps/Wps}$;$Tyr^c/Tyr^c$;$Rpe65^{+/+}$ mice. For genotyping, the *Kit* gene was amplified with Kit-F and Kit-R, and the Rpe65 gene was amplified with Rpe65-F5 and Rpe65-R (*Supplementary file 1*: Key Resources Table) as previously described (*Guo et al., 2010*; *Wenzel et al., 2001*). *rd1* (*Pde6b^{rd1}*, FVB genetic background) and *rd10* (*Pde6b^{rd10}*, C57BL/6J genetic background) mice were purchased from Vital River Laboratory (Beijing, China) and Jackson Laboratory. All mice were fed a standard laboratory chow and maintained at 21–23°C with a 12 h-light/12 hr-dark photoperiod.

Enucleated eyes from euthanized mice were fixed in 4% paraformaldehyde (PFA) overnight at RT and paraffin-embedded according to standard procedures. Paraffin sections (5 μm thickness) were prepared using a Leica microtome, deparaffinized by immersion in Xylene, and rehydrated. The sections were stained with hematoxylin for 5 min and eosin for 3 min and mounted using resinene. The thickness of $Kit^{+/+}$ and $Kit^{Wps/Wps}$ retinas was measured in the dorsal central region at a distance between 300 and 700 μm from the optic nerve head.

### In vivo light injury and RNA-seq analysis

For light-induced retinal damage, 2-month-old albino mice were first dark-adapted for 12 hr and then exposed to LED light (15,000 lux) in non-reflective cages for the indicated number of days. For $Kit^{+/+}$ and $Kit^{Wps/Wps}$ mice, 3-month-old mice were first dark-adapted for one day and then exposed to LED light (15,000 lux) in non-reflective cages for the indicated number of days. During light exposure, mice were allowed free access to food and water. For ERG measurements, mice were dark-adapted for 1 day after light exposure.

For RNA-seq analysis, total RNAs of neural retinas were extracted using TRIzol reagent (Invitrogen) after 1 day of light exposure following the time frame shown in *Figure 1A*. Purified RNA samples were used to generate RNA libraries for HiSeq 4000 (PE150 sequence). Experiments and data normalization were performed by Beijing Novogene (China). The raw data of RNA seq were deposited to the GEO database (GSE146176) and are available online (https://www.ncbi.nlm.nih.gov/geo/query/acc.cgi?acc=GSE146176).

## Electroretinography

Retinal function was evaluated by electroretinography (ERG) as previously described (*Li et al., 2020*). Briefly, mice were dark-adapted overnight and anesthetized with a mixture of ketamine and xylazine. The pupils were dilated and the corneas were anesthetized with atropine sulfate and proparacaine hydrochloride. Then a silver loop electrode was placed over the cornea to record the ERGs, while needle reference and ground electrodes were inserted into the cheek and tail, respectively. Mice were stimulated by flash light varying in intensity from $-5.0$ to 35 log scotopic candlepower-sec/$m^2$ in a Ganzfeld dome (Roland Q400, Wiesbaden, Germany). For light-adapted ERG recordings, a background light of 30 cd/$m^2$ was applied for 5–10 min to suppress rod responses. The stimulus light intensity was attenuated with neutral density filters (Kodak, Rochester, NY) and luminance was calibrated with an IL-1700 integrating radiometer/photometer (International Light, Newburyport, MA).

## Cell cultures, light damage, siRNA and transfection

The 661W photoreceptor cell line was generously provided by Dr. Muayyad Al-Ubaidi (Department of Cell Biology, University of Oklahoma Health Sciences Center, Oklahoma City, OK). The cells were routinely cultured in Dulbecco's modified Eagle's medium (Gibco) supplemented with 10% FBS and 1% penicillin/streptomycin and kept at 37°C in a humidified atmosphere with 5% $CO_2$. 661 W cell line was tested negative for mycoplasma contamination by the MycoProbe Mycoplasma detection Kit (Cat# CUL001B, R&D) and positive for photoreceptor cell marker Recoverin and Opsin by IF.

For LD, 661W photoreceptor cells were kept at 37°C in a humidified atmosphere with 5% CO2 and exposed to high-intensity light (9000 lux) for 8 hr. Then the cells were subjected to cell count analysis and PI staining (1:200, Sigma).

For siRNA knockdown, specific siRNAs for mouse *Nfe2l2*, *Hmox* or *Kitl* and a negative control siRNA were designed and produced by Gene-Pharma (GenePharma, China). 20 nM si- *Nfe2l2*, si-*Hmox1* or si-*Kitl* was transfected into 661 W cells at 40% confluency in culture dishes using LipoJetin vitro transfection Kit (SignaGen Laboratories) according to the manufacturer's protocol. An equivalent amount of control siRNA (si–C) was used as a negative control. The sequences for all siRNAs are shown in *Supplementary file 1*: Key Resources Table.

For DNA transfections, plasmids of pCDNA3.1-encoding either KIT$^{WT}$ cDNA or KIT$^{Wps}$ cDNA were from Dr. Xiang Gao (Nanjing University). 661 W cells were seeded in six-well plates at $7.5 \times 10^4$ cells/well before transfection. When the cell density reached 30% confluence, cells were transfected with appropriate dilutions of 2 µg plasmid DNA using PolyJet reagent (SignaGen) at the indicated concentrations following the company's instructions.

## Western blots and assays of protein phosphorylation

For analysis of phosphorylation of KIT, mice were anesthetized with a mixture of ketamine and xylazine, and 1 µl of KITL (5.6 nM) was intravitreally injected into the eye. One hour thereafter, neural retinas were isolated and subjected to western blot analysis. For protein extraction, retinas were isolated in DMEM and transferred to 1.5 ml centrifuge tubes containing a mixture (100 µl) of membrane protein lysis solution (Byotime, China) and protease inhibitors (Cocktail Set I, Calbiochem). Retinas were lysed on ice for 10 min with a Micro Tissue Grinder. Samples were separated by 8% SDS-PAGE and the blotting process was previously described (*Li et al., 2017*). Information of antibodies used in western blot is shown in *Supplementary file 1*: Key Resources Table. These antibodies were used at 4°C overnight. The quantification of the protein bands was done using the software of Image J.

For isolated RPE and photoreceptor tissue, RPE sheets were isolated from 2-month-old C57BL/6J mice as described previously (*Fernandez-Godino et al., 2016*) and then subjected to western blotting analysis. The neural retinas were isolated in Hanks solution and then freshly frozen in OCT

compound. The photoreceptor layer was at the bottom of the embedding block. The first cryosection of neural retina (30 µm) was transferred to 1.5 ml centrifuge tubes containing a mixture (30 µl) of membrane protein lysis solution and then subjected to western blotting analysis as described above.

For cell protein extraction, cells were washed with 1 × PBS and then lysed by membrane protein lysis solution on ice. In addition, we used cell fractionation kit-standard (Abcam, ab109719) to extract cytoplasmic and nuclear proteins and performed the analysis according to the standard protocols.

## RT-PCR and quantitative RT- PCR

Total neural retinal RNA or total RPE RNA was extracted by Trizol reagent (Invitrogen) according to the manufacturer's protocol. Total RNA for RT-PCR was converted to cDNA with a reverse transcriptase kit and random primers (Promega), according to the manufacturer's manual. PCR products were size-fractionated by 2% agarose gel electrophoresis. The procedure for quantitative RT-PCR has been described (*Li et al., 2017*). Sequences of primers used in this study are shown in *Supplementary file 1*: Key Resources Table.

## Immunohistochemistry and TUNEL assay

Information on the antibodies used in immunofluorescence (IF) is shown in *Supplementary file 1*: Key Resources Table. For immunoassays of KITL, KIT, EAAT1, Flag and HMOX1, eyes were freshly embedded in OCT compound and snap frozen. Cryosections (14 µm) were post-fixed in ice-cold acetone for 15 min and permeabilized with 0.1% Triton-X-100 for 10 min. Rat anti-KIT (ACK45), goat anti-EAAT1, rabbit anti-Flag, goat anti-KITL and rabbit anti-HMOX1 antibodies were used at 4℃ overnight and Alexa 488-conjugated donkey anti-rat, anti-goat or anti-rabbit antibodies were used at RT for 1 hr. The sections were examined and photographed with a Zeiss fluorescence microscope.

For other immunoassays, the corneas were penetrated with a needle in 1 × PBS and the eyes fixed in 4% PFA for 1 hr at RT. The sections (5 µm) were heated in boiled citrate buffer for 2 min for epitope retrieval. Primary antibodies were: mouse anti-rhodopsin, rabbit anti-opsin, mouse anti-PKCα, rabbit anti-OTX2, goat anti-KITL, rabbit anti-KIT (AB5506), rabbit anti NRF2 and rabbit anti-GFAP. The antibodies were all used at RT for 2 hr. The primary antibodies were revealed with Alexa 594-conjugated donkey anti-mouse or anti-rabbit and Alexa 488-conjugated donkey anti-goat or anti-mouse antibodies at RT for 1 hr.

Cultured cells were fixed in 4% PFA for 25 min at RT and permeabilized with 0.1% Triton X-100. 5% BSA was used to block the samples at RT for 30 min. Rat anti-KIT (ACK2) antibody was used at RT for 2 hr.

For TUNEL assays, in situ cell death detection kit (Roche, Mannheim, Germany) was used according to the manufacturer's protocol.

## AAV8 vector construction and virus injection

For the AAV8-CMV-KITL vector, the vector of pcDNA3.0-*Kitl* was a gift from Dr. Bernhard Wehrle-Haller (University of Geneva, Switzerland). The coding sequence of the mouse *Kitl* was cloned from the vector and inserted into the AAV2/8-CMV-MCS empty vector (Genechem, China). For the AAV8-RPE65-KITL-T2A-Flag vector, GGCAGCGGCGAGGGCAGAGGAAGTCTTCTAACATGCGGTGACG TGGAGGTCCCGGCCCT of the T2A the sequence was linked to the coding sequence of *Kitl* at the 3' terminus and then inserted into the AAV2/8-RPE65-MCS-Flag vector (Genechem, China). For AAV8-RHO-KITL and AAV8-RHO-Flag vector, we first replaced the CMV promoter sequence of the AAV2/8-CMV-MCS vector with the Human RHO promoter sequence (GeneBank accession number: U16824) to obtain the AAV2/8-RHO-MCS vector according to previous report (*Allocca et al., 2007*) and then inserted the coding sequence of the mouse *Kitl* or Flag into the AAV2/8-RHO-MSC empty vector. For AAV8-CMV-Hmox1 vector, the coding sequence of the mouse *Hmox1* gene was amplified by reverse transcription PCR using the primers for Hmox1-F and Hmox1-R as shown in *Supplementary file 1*. The full-length cDNA of *Hmox1* was cloned from *Kit*$^{+/+}$ retinal total cDNA and inserted into the AAV2/8-CMV-MCS empty vector (Genechem, China) via *Age*I restriction sites introduced by the PCR reaction. The AAV8-CMV-GFP vector was purchased from Genechem Company (Shanghai, China).

Virus was injected into the subretinal space as previously described (*Xiong et al., 2015*). Approximately 0.5 µl of AAV8 supernatant ($2.2 \times 10^{12}$ genome copies/ml) was introduced into the subretinal space of 2- to 3-month-old mice using a pulled angled glass pipette under direct observation aided by a dissecting microscope under dim light. Following injections, 1% atropine eye drops, tetracycline, and cortisone acetate eye ointments were applied. For subretinal injection, neonatal *rd1* and *rd10* mice were kept on ice for 2 min and then subretinally injected with approximately 0.3 µl of AAV8 using micro-injection glass pipettes.

## In situ hybridization

Retinas from 2-month-old albino mice kept under normal light or high intensity light for 1 day were collected in PBS and fixed overnight at 4°C using 4% paraformaldehyde (PFA). In situ hybridization was performed according to previous report (*Chen et al., 2019*). The *Kitl* riboprobes was obtained using pcDNA3.0-*Kitl* (gift from Dr. Bernhard Wehrle-Haller).

## Statistical analysis

Data are from at least three replicates for each experimental condition and are represented as mean ± standard deviation (SD). Analysis of variance (ANOVA) was used to determine the significance between population means. Student's t-test was used to determine the significance of differences when only two groups were compared. $p < 0.05$ was considered to be statistically significant. Significant differences between groups are noted by *or **.

## Acknowledgements

We thank Drs. Xiang Gao, Wei Li for providing the *Kit^{Wps}* mice and reagents, Dr. Bernie Wehrle-Haller for providing reagents, Drs. Muayyad R Al-Ubaidi and Feng Gu for 661 W cell line, and Drs. Heinz Arnheiter, Wenjun Xiong, and Wei Li (NEI) for thoughtful comments and editing of the manuscript.

## Additional information

### Funding

| Funder | Grant reference number | Author |
|---|---|---|
| National Natural Science Foundation of China | 81800838 | Huirong Li |
| National Natural Science Foundation of China | 81770946 | Ling Hou |

The funders had no role in study design, data collection and interpretation, or the decision to submit the work for publication.

### Author contributions

Huirong Li, Conceptualization, Data curation, Formal analysis, Funding acquisition, Investigation, Writing - original draft, Writing - review and editing; Lili Lian, Bo Liu, Data curation, Formal analysis, Investigation, Writing - review and editing; Yu Chen, Jinglei Yang, Shuhui Jian, Jiajia Zhou, Investigation, Writing - review and editing; Ying Xu, Resources, Writing - review and editing; Xiaoyin Ma, Resources, Funding acquisition, Writing - review and editing; Jia Qu, Conceptualization, Resources, Supervision, Project administration, Writing - review and editing; Ling Hou, Conceptualization, Resources, Data curation, Formal analysis, Supervision, Funding acquisition, Writing - original draft, Project administration, Writing - review and editing

### Author ORCIDs

Huirong Li  https://orcid.org/0000-0002-6631-5553
Ling Hou  https://orcid.org/0000-0003-0705-8099

## Ethics

Animal experimentation: All animal experiments were carried out in accordance with the approved guidelines of the Wenzhou Medical University Institutional Animal Care and Use Committee (Permit Number: WZMCOPT-090316).

## Decision letter and Author response

Decision letter https://doi.org/10.7554/eLife.51698.sa1
Author response https://doi.org/10.7554/eLife.51698.sa2

# Additional files

## Supplementary files

• Supplementary file 1. Key resources table.

• Transparent reporting form

## Data availability

Sequencing data have been deposited in GEO under accession codes 146176. All data generated or analysed during this study are included in the manuscript and supporting files. Source data files have been provided for Figure 1—figure supplement 1, Figure 2, Figure 3, Figure 4, Figure 5, Figure 5—figure supplement 1, Figure 8, Figure 9, Figure 9—figure supplement 1.

The following dataset was generated:

| Author(s) | Year | Dataset title | Dataset URL | Database and Identifier |
|---|---|---|---|---|
| Li H,  Hou L | 2020 | Next generation sequencing facilitates quantitative analysis of retinal transcriptomes under normal light and high-intensity light condition | https://www.ncbi.nlm.nih.gov/geo/query/acc.cgi?acc=gse146176 | NCBI Gene Expression Omnibus, GSE146176 |

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
