## [Decision Letter]

**Acceptance summary:**

Your paper is significant as it reports the discovery that activation of the *Kitl*/SCF pathway is an early stress response to photoreceptor degeneration, upstream of NRF2/HMOX1. The fact that it is providing protection to light damage makes it clinically relevant and this should be of interest to the readership of *eLife*.

**Decision letter after peer review:**

Thank you for submitting your article "KIT ligand protects against both light-induced and genetic photoreceptor degeneration" for consideration by *eLife*. Your article has been reviewed by three peer reviewers, and the evaluation has been overseen by a Reviewing Editor and Marianne Bronner as the Senior Editor. The reviewers have opted to remain anonymous.

The reviewers have discussed the reviews with one another and the Reviewing Editor has drafted this decision to help you prepare a revised submission.

The three reviewers appreciated the discovery that activation of the *Kitl*/SCF pathway is an early stress response to photoreceptor degeneration, upstream of NRF2/HMOX1, and that it is providing protection to light damage. The clinical relevant of this observation is therefore very high.

However, the reviewers noticed a number of important points that must be addressed before the paper can be considered for publication:

1) First, you must show that the effect in autonomous and not driven by the RPE where. Using cell specific promoters to rescue the phenotype would allow you to address this point. The three reviewers were concerned about this point that must absolutely be addressed.

2) One reviewer was also concerned with the use of the 661W tumor cell line that might not be appropriate for these experiments. Have you used any other cell line? In any case, this must be discussed in the paper.

3) There were also concerns about the quality of the Western blots, although the reviewers believe that, considering the difficulty to perform these experiments on retinal extract, they might not be perfect. However, you should discuss the specificity of the antibodies and try to show better images. The specificity of these antibodies might have been partially demonstrated by the blots of AAV-mediated KITL or HMOX1 overexpression in the eye and transfection of KIT^WT^ or *Kit^Wps^* in the cell line.

4) The problem appears more serious with the IF data and these must be improved to convince the reviewers that what you are trying to demonstrate. In particular, the RPE must absolutely be included in the images.

5) As mentioned above, the cell specificity of the AAV virus must be better demonstrated as cell autonomy is such an important point of the paper.

6) The ERGs might also need to be improved.

Reviewer #1:

The *Kitl/Kit* pathway has not been well studied in the eye. The findings of this group that activation of the *Kitl/Kit* pathway is one of the early stress responses to photoreceptor degeneration and that it is the upstream of NRF2-dependent HMOX1 upregulation are novel. The authors provided strong evidence that the *Kitl/Kit* pathway is necessary and sufficient to provide protection to photoreceptors following light damage. Further study using genetic retinal degeneration mouse models indicated a generic protective effect of that *Kitl/kit* pathway, which may have impact on RP treatment. Overall, I found that this manuscript delivers high quality data and is clearly written. The results are compelling and have appropriate quantifications whenever possible.

One part of the story that is less clear to me is whether the activation of the KITL/KIT pathway in the photoreceptors is in a cell-autonomous manner or not. The authors suggested that it is the photoreceptors upregulating *kitl* following light damage or AAV-KITL infection. What is the expression level of *kitl* and kit in the RPE vs. retina in uninjured eyes? And have the authors examined the *kitl* expression in the RPE following light damage by qPCR or immunostaining (Figure 1I used retinal section without RPE)? In terms of the rescue experiment by viral vectors, AAV-CMV vector also drives strong transgene expression in the RPE cells, which is evident in Figure 4—figure supplements 1 and 2. The strong staining of *Kitl* in the OS is likely from the cilia of the RPE cells. This is also obvious in Figure 5F, where KITL and HMOX1 overexpression are not overlapping. What the authors can do in the future is to use RPE-specific or photoreceptor-specific promoters to drive *Kitl* expression and examine the rescue effects. As for this manuscript, I think that the authors should discuss the possible involvement of RPE and/or the other cell types in the retina in the activation of the KITL/KIT pathway.

Reviewer #2:

The manuscript by Li et al. focuses on the role of KIT/KITL signaling in photoreceptors of the retina as it relates to response to light damage. Light damage in a sensitive mouse strain leads to increased signaling through the KITL/KIT/MAPK pathway. They used a Kit hypomorphic allele on pigmented and albino mouse strains to show that signaling through this pathway is necessary to protect photoreceptors. In addition, AAV delivery was used to demonstrate that ectopic KITL expression could partially protect photoreceptors using their light damage model. Additional experiments were focused on elucidating the other components of the pathway such as HMOX1 and NRF2. Those studies were primarily ectopic expression and knockdown experiments. The authors used in vivo mouse models but also a cell line from a mouse tumor (661W). The strengths of the manuscript include the emphasis on in vivo studies and the potential impact on a major human disease. The limitations are scientific rigor, use of the retinoblastoma 661W cell line and incomplete experimental details. Specific comments include:

1) Throughout the manuscript, the western blots are not convincing and data to demonstrate specificity of the antibodies should be provided.

2) Throughout the manuscript, the immunofluorescence data are not convincing. The photoreceptor inner segments are a common site of non-specific staining and efforts should be made to demonstrate specificity and localization of the signal. For example, it is impossible to distinguish inner segments from Muller apical microvilli in their images.

3) Controls should be included to definitively demonstrate which cells are infected with the AAV vector. AAV8 should infect all cell types and they did not provide convincing data on photoreceptor specificity. This is important because cell autonomous versus non-autonomous signaling is an important distinction for KITL/KIT signaling.

4) The 661W cell line is a mouse retinoblastoma with features of multiple cell types and this is not an appropriate proxy for normal cells.

5) The in vivo rescue was not well controlled. How many mice were used, how long were they followed, empty vector was not used in all experiments as should have been done.

Reviewer #3:

The authors present a detailed report exploring novel genetic factors associated with light induced damage of photoreceptor cells. The authors identify a gene previously unassociated with light damage, KITL (KIT Ligand, also known as SCF, SLF), as upregulated following light damage. This leads to activation of its receptor, KIT (also known as CD117, c-KIT), and down-stream signaling pathways. Inactivation of the KIT pathway using a genetic mutant results in worsening of light induced photoreceptor loss. Over-expression of KITL in photoreceptors via AAV transduction can protect against light induced damage, but not in animals lacking functioning KIT. The authors identify HMOX1 as a potential downstream target of KIT signaling as a factor important for mediating this protective effect. They show that activation of the KIT pathway with either light damage or over-expression of KITL results in upregulation of HMOX1. This effect on HMOX1 appears to be regulated through nuclear translocation of NRF2 which is inhibited in KIT deficient animals. The authors then show that over-expression of HMOX1 via AAV transduction is sufficient to partially protect photoreceptors from light induced damage. Finally, the authors show that activation of the KIT signaling pathway through over-expression of KITL is capable of protecting photoreceptors in two models of retinitis pigmentosa.

Overall the manuscript is well written with strong supporting evidence for the claims stated. The experiments are presented in a logical order with convincing data. I would recommend this manuscript for publication after revision.

I highly suggest giving the manuscript an additional editing pass for spelling and grammar as well as wording of some sentences (e.g. The sentence "We observed that after 2 weeks of injection, high-level KITL expression was restricted to photoreceptor cells and the RPE…" implies that the injections were performed for 2 weeks, when in reality they were done once and then the animals were followed up 2 weeks later. There are numerous occasions of this type of phrasing and other similar examples that are confusing and detract from the overall manuscript).

I would also ask that the authors revisit their Discussion and explore more rationale as to the protective effect of KIT signaling on photoreceptors. The authors mainly restate their results and do not spend anytime analyzing how HMOX1 could be mediating its protective effect and this discussion is necessary to explain how KIT signaling can be protective. The authors also need to address alternative pathways downstream of KIT that may also be contributing to photoreceptor protection. The authors admit that HMOX1 is only capable of partially rescuing photoreceptors. Are there other pathways which may be of interest to examine next?

There are several ERG graphs that show only noisy traces which are not useful for assaying retinal function (Figure 3B, Kit+/+ no light damage photopic; Figure 8F, AAV8-KITL transduced *rd10* homozygous animals; Figure 9—figure supplement 1B, both traces). If these panels are representative of all data collected for these experiments then they would either need to be repeated or omitted from the manuscript. The associated morphological data is sufficient to show a protective effect of KITL/damaging effect of removing KIT signaling without needing ERG data to show functional deficits.

Is the AAV8 construct used for HMOX1 identical to the one used for KITL (besides the gene)? If so, why is there such broad expression of HMOX1 compared to KITL after transduction? The promoter used is CMV so one would expect broad expression in most cells of the retina.

KIT appears to be mainly expressed on the inner segments of photoreceptors while KITL appears to be mainly expressed on the outer segments of photoreceptors/RPE villi. Do photoreceptors normally express KITL or is this exogenously supplied by RPE during stress? Could the authors provide an in situ hybridization showing if KITL is expressed in photoreceptors or in RPE (or both)?

---

## [Author Response]

However, the reviewers noticed a number of important points that must be addressed before the paper can be considered for publication:1) First, you must show that the effect in autonomous and not driven by the RPE where. Using cell specific promoters to rescue the phenotype would allow you to address this point. The three reviewers were concerned about this point that must absolutely be addressed.

We appreciate your and your reviewers’ comments. According to your suggestion, we have performed additional experiments and provide stronger evidence supporting the conclusion that photoreceptor cells upregulate KITL expression to prevent photoreceptor degeneration during light damage. Firstly, based on the results of gene expression analysis by qPCR, WB, IF and ISH, we show that photoreceptor cells upregulate *Kitl* expression in response to light damage (see new Figure 1I, Figure 1—figure supplement 2 and Figure 1—figure supplement 3). The details are described in the revised Results section (subsection “Light damage upregulates endogenous KITL in photoreceptor cells”, last paragraph). Secondly, we expressed KITL specifically in photoreceptor cells or RPE cells, using AAV8 viruses allowing for KITL expression either under the control of the photoreceptor cell-specific promoter RHO (AAV8-RHO-KITL) or the RPE cell-specific promoter RPE65 (AAV8-RPE65-KITL), and analyzed its effect on LD (see new Figure 5，Figure 5—figure supplement 1). The results are consistent with the view that it is predominantly KITL generated in photoreceptors and not KITL generated in RPE cells that protects photoreceptors against LD. In other words, KITL acts in an autocrine and not paracrine fashion. The results are described in the subsection “Overexpression of KITL in photoreceptor cells prevents light-induced retinal degeneration”.

2) One reviewer was also concerned with the use of the 661W tumor cell line that might not be appropriate for these experiments. Have you used any other cell line? In any case, this must be discussed in the paper.

We appreciate your suggestions and understand the reviewer’s concerns. As far as we know, the 661W cell line is the only cell line representing photoreceptor cells, and it has been widely used for investigations of photoreceptor cell biology and function (Tan et al., 2004). Recently, Wheway et al. have performed whole transcriptome RNA sequencing of 661W cells and showed that 661W cells originated from photoreceptor cells and can be used as a cell model for studying retinal ciliopathies (Wheway et al., 2019). We, therefore, reasoned that 661W cells serve as an appropriate model of photoreceptor cells for in vitro studies, designed principally to support our main in vivo results. According to your suggestion, we have modified the original sentence in the Results section (subsection “KIT signaling regulates *Hmox1* expression in photoreceptor cells”, second paragraph). To further address your concern, we also isolated primary photoreceptor cells from C57BL/6J mice at P7 to examine the relationship between KITL/KIT and *Hmox1* (Author response image 1). The IF data show that a high proportion of the primary retinal cells expressed photoreceptor specific marker Rhodopsin (Author response image 1), suggesting they are photoreceptor cells. In response to stimulation of KITL, the primary cells not only upregulated phosphorylated KIT (Author response image 1), but also turned on *Hmox1* expression (Author response image 1). These results are consistent with the studies in 661W cell line. However, we do not think that addition of this figure would add more relevant information to the main message of our manuscript.

**Author response image 1. respfig1:** KITL induces *Hmox1* expression in primary retinal cells. (**A**) Representative immunostaining images of anti-Rhodopsin in cultured retinal cells from C57BL/6J mice at P7. (**B**) Western blots show the levels of phosphorylated KIT after stimulation with KITL in the primary cells. (C, D) The expression analysis of *Hmox1* in the primary cells after stimulation with KITL through RT-PCR (**C**) and qPCR (**D**) examination (n=3). * or ** indicates *p*<0.05 or *p*<0.01. Bar: 50 μm.

3) There were also concerns about the quality of the Western blots, although the reviewers believe that, considering the difficulty to perform these experiments on retinal extract, they might not be perfect. However, you should discuss the specificity of the antibodies and try to show better images. The specificity of these antibodies might have been partially demonstrated by the blots of AAV-mediated KITL or HMOX1 overexpression in the eye and transfection of KIT^WT^ or Kit^Wps^ in the cell line.

We appreciate your and your reviewers’ suggestions. We have further analyzed the specificity of the anti-KITL antibody through overexpression of KITL in 661W cell line by AAV8-KITL and human recombinant KITL peptides (Author response image 2). The IF and WB data consistently show that the antibody is specific to KITL (Author response image 2). However, we do not think that addition of this figure would add more relevant information to the main message of our manuscript. In addition, the data of IF and WB from the retinas infected by AAV8-RPE65-KITL-T2A-Flag virus also showed that the KITL antibody can specifically recognize the KITL protein in the retina (see new Figure 5C,D). The specificity of other antibodies can also be seen by which tissue is stained (see new Figure 1—figure supplement 3B). For instance, MITF is known to be an RPE-specific marker and Rhodopsin a photoreceptor cell specific marker. Furthermore, overexpression in vivo or in vitro serves as quality control for the KIT and HMOX1 antibodies (see Figure 6—figure supplement 1 and Figure 8A, B). To improve the quality of the western blots, we repeated most of the corresponding experiments and replaced previous images with improved ones (Figure 2I; Figure 2—figure supplement 1A; Figure 4A; Figure 6C, E; Figure 7A; Figure 8A).

**Author response image 2. respfig2:** Analysis of the anti-KITL antibody in photoreceptor cells. (**A**) Immunostaining images of anti-KITL in 661W photoreceptor cells infected with AAV8 (left panel) or AAV8-KITL (right panel) virus. (**B**) Western blots show the specific recognition of the anti-KITL antibody to Human recombinant KITL protein (18 kDa). (**C**) Immunostaining images of Goat anti-IgG (left panels) and Goat anti-KITL (right panels) in the retinas from 2-month-old albino mice kept under normal light or LD (15, 000 Lux) condition for 1 day. GCL, ganglion cell layer; INL, inner nuclear layer; ONL, outer nuclear layer. Bar: 50 μm.

4) The problem appears more serious with the IF data and these must be improved to convince the reviewers that what you are trying to demonstrate. In particular, the RPE must absolutely be included in the images.

We appreciate your and your reviewers’ suggestions. For KITL expression, we have repeated the IF experiments of anti-KITL in LD treated retinas and provide new improved images that include photoreceptor cells and RPE (see new Figure 1I). To further analyze the specificity of the anti-KITL antibody, we used IgG antibody as a negative control (Author response image 2). To avoid non-specific staining in the photoreceptor inner segment, we isolated photoreceptor tissue from frozen sections as shown in Figure 1—figure supplement 3A and then used western blotting to detect KITL by anti-KITL antibody (see Figure 1—figure supplement 3B, C). In addition, *Kitl* expression in photoreceptor cells was also confirmed by the data of in situ hybridization using a *Kitl* probe (see Figure 1—figure supplement 3D). All of these data support the notion that KITL is indeed expressed in photoreceptor cells.

For KIT expression, we also repeated the IF experiment with anti-KIT (AB5506) in the retina and replaced the original image with a new one (see new Figure 2—figure supplement 1B). To analyze whether KIT immunoreactivity signal in the photoreceptor layer is from Müller glia cells, we performed double immunostaining with anti-KIT antibody (ACK45) and anti-EAAT1 antibody (Müller glia-specific marker) in C57BL/6J mice and *rd10/rd10* mice (whose photoreceptor cells are vastly diminished) at postnatal 2 month (see Figure 2—figure supplement 2). The results showed that KIT immunoreactivity in the photoreceptor layer is unlikely to come from interdigitating apical microvilli of Müller glial cells. The results are described in the second paragraph of the subsection “The *Kit^Wps^* mutation does not cause detectable changes in retinal function and structure but disrupts KIT activation in the retina”.

5) As mentioned above, the cell specificity of the AAV virus must be better demonstrated as cell autonomy is such an important point of the paper.

We appreciate your suggestion and have performed corresponding experiments as described above in the response to Editor comment 1. The data (see new Figure 5 and Figure 5—figure supplement 1) indicate that overexpression of KITL in photoreceptor cells prevents light-induced retinal degeneration (subsection “Overexpression of KITL in photoreceptor cells prevents light-induced retinal degeneration”).

6) The ERGs might also need to be improved.

We have repeated the ERG examination and replaced previous images with improved images (Figure 3B; Figure 9F; Figure 9—figure supplement 1B) in this revision.

Reviewer #1:[…] One part of the story that is less clear to me is whether the activation of the KITL/KIT pathway in the photoreceptors is in a cell-autonomous manner or not. The authors suggested that it is the photoreceptors upregulating kitl following light damage or AAV-Kitl infection. What is the expression level of kitl and kit in the RPE vs. retina in uninjured eyes? And have the authors examined the kitl expression in the RPE following light damage by qPCR or immunostaining (Figure 1I used retinal section without RPE)?

We appreciate your comments. According to your suggestions, we have examined KITL expression in RPE and photoreceptor cells of light-injured retinas or uninjured retinas. The gene expression analysis, including q-PCR, IF, WB and ISH (see new Figure 1I, Figure 1—figure supplement 2 and Figure 1—figure supplement 3), consistently showed that photoreceptor cells are able to upregulate *Kitl* expression in response to light stress (subsection “Light damage upregulates endogenous KITL in photoreceptor cells”, last paragraph).

In terms of the rescue experiment by viral vectors, AAV-CMV vector also drives strong transgene expression in the RPE cells, which is evident in Figure 4—figure supplements 1 and 2. The strong staining of Kitl in the OS is likely from the cilia of the RPE cells. This is also obvious in Figure 5F, where KITL and HMOX1 overexpression are not overlapping. What the authors can do in the future is to use RPE-specific or photoreceptor-specific promoters to drive Kitl expression and examine the rescue effects.

We appreciate your great comment. According to your suggestions, we now utilized the photoreceptor cell-specific promoter RHO (AAV8-RHO-KITL) and RPE cell-specific promoter RPE65 (AAV8-RPE65-KITL) to drive KITL expression separately in the respective cell types and then analyzed the effects on LIRD (see new Figure 5, Figure 5—figure supplement 1 and the response as described above in the Editor comment 1). Indeed, KITL exerts its protective role when expressed in photoreceptors and not when expressed in the RPE. The results are described in the subsection “Overexpression of KITL in photoreceptor cells prevents light-induced retinal degeneration”.

As for this manuscript, I think that the authors should discuss the possible involvement of RPE and/or the other cell types in the retina in the activation of the KITL/KIT pathway.

We appreciate your suggestions. We now discuss this point in the Discussion section of the revision (fourth paragraph).

Reviewer #2:1) Throughout the manuscript, the western blots are not convincing and data to demonstrate specificity of the antibodies should be provided.

We appreciate your suggestions and have further analyzed the specificity of the anti-KITL antibody through overexpression of KITL in 661W cell line by AAV8-KITL and the human recombinant KITL peptides as you can see from Author response image 2 and the response as described above to Editor comment 3. In addition, the data of IF and WB from the retinas infected by AAV8-RPE65-KITL-T2A-Flag virus also showed that the KITL antibody specifically recognizes KITL protein in the retina (see new Figure 5C, D). As mentioned above, the specificity of other antibodies can also be seen by which tissue is stained (see new Figure 1—figure supplement 3B). For instance, MITF is known to be an RPE-specific marker and Rhodopsin a photoreceptor cell specific marker.

Furthermore, overexpression in vivo or in vitro serves as quality control for the KIT and HMOX1 antibodies (see Figure 6—figure supplement 1 and Figure 8A, B). To improve the quality of the western blots, we repeated most of them and replaced previous images with improved images (Figure 2I; Figure 2—figure supplement 1A; Figure 4A; Figure 6C, E; Figure 7A; Figure 8A).

2) Throughout the manuscript, the immunofluorescence data are not convincing. The photoreceptor inner segments are a common site of non-specific staining and efforts should be made to demonstrate specificity and localization of the signal. For example, it is impossible to distinguish inner segments from Muller apical microvilli in their images.

We appreciate your suggestions. For KITL expression, we have repeated the IF experiment of anti-KITL in LD treated retinas and replaced the former figure with a new and improved image which includes photoreceptor cells and RPE (Figure 1I). To further analyze the specificity of the anti-KITL antibody in the retina, we used IgG antibody as a negative control. The IF data showed that the immunoreactivity signal of anti-KITL antibody was increased in the photoreceptor cells after LD treatment, while the immunoreactivity signal of anti-IgG antibody was not observed in the photoreceptor cells (Author response image 2 and response to Editor comment 3 described above). To avoid the non-specific staining in the photoreceptor inner segment by IF, we isolated photoreceptor tissue from frozen sections as shown in Figure 1—figure supplement 3A and used western blotting to examine KITL using the anti-KITL antibody. The western blots showed that the level of KITL protein is much higher in the photoreceptor cells than in RPE cells (Figure 1—figure supplement 3B, C). In addition, *Kitl* expression in photoreceptor cells was confirmed by in situ hybridization using a Kitl probe (Figure 1—figure supplement 3D). These data indicate that KITL is expressed in photoreceptor cells.

For KIT expression, we also repeated the IF experiment using anti-KIT (AB5506) on the retina and replaced the original image with an improved one (Figure 2—figure supplement 1B). To analyze whether KIT immunoreactivity signal in the photoreceptor layer is from Müller glia cells, we performed double immunostaining with anti-KIT antibody (ACK45) and anti-EAAT1 antibody (Müller glia-specific marker) in C57BL/6J mice and *rd10/rd10* mice with their vastly diminished photoreceptor cells at postnatal 2 month (see Figure 2—figure supplement 2). The results showed that the KIT immunoreactivity in the photoreceptor layer is unlikely to come from interdigitating apical microvilli of Müller glial cells. The results are described in detail in the Results section (subsection “The *Kit^Wps^* mutation does not cause detectable changes in retinal function and structure but disrupts KIT activation in the retina”, second paragraph).

3) Controls should be included to definitively demonstrate which cells are infected with the AAV vector. AAV8 should infect all cell types and they did not provide convincing data on photoreceptor specificity. This is important because cell autonomous versus non-autonomous signaling is an important distinction for KITL/KIT signaling.

We appreciate your suggestions. For control groups, we first analyzed the effect of AAV8-CMV empty vector or AAV8-CMV-Flag on light-induced photoreceptor degeneration. The 2-month-old albino mice were infected by the virus for 2 weeks and then exposed to NL or high-intensity light for 3 days. At 2 weeks post-infection, IF data showed that AAV8-CMV-Flag virus successfully drove Flag expression in photoreceptor cells and RPE. After LD of 3 days, the photoreceptor degeneration in the retinas infected by AAV8-CMV-Flag was similar to that in the retinas infected by the AAV8-CMV-empty, suggesting that AAV8-CMV-empty can be used as an appropriate control virus (Author response image 3). Due to the fact that AAV8-CMV-empty has already been used as a control virus in the original version, we still used AAV8-CMV empty virus as a control virus in *Kit^Wps/Wps^* albino mice kept under NL or high-intensity light condition in this revision. In addition, we also injected AAV8-CMV empty virus into the eyes of *rd1* and *rd10* mice as control groups and replaced all uninjected groups with the injection of AAV8-CMV empty virus in this revision (Figure 4H-J; Figure 9; Figure 9—figure supplement 1).

**Author response image 3. respfig3:** The effect of AAV8-Flag virus on the LIRD is similar to that of AAV8 empty virus. (**A**) Immunostaining images of anti-Flag in the retinas from 2-month-old albino mice 2 weeks after intraocular injection of AAV8 or AAV8-Flag virus. (B, C) Histological analysis ofretinas from the albino mice infected with AAV8 or AAV8-Flag and kept under NL or LD (15, 000 Lux) condition for 3 days (n=5) (**B**) and the curve diagram (**C**) showing the thickness of ONL fromthe *albino* retinas under the indicated conditions. GCL, ganglion cell layer; INL, inner nuclear layer; ONL, outer nuclear layer; IS, inner segment; OS, outer segment. ** indicates *p*<0.01. Bar: 50 μm.

For the cell specificity of KITL overexpression, according to your suggestion, we utilized photoreceptor cell specific promoter RHO (AAV8-RHO-KITL) or RPE cell specific promoter RPE65 (AAV8-RPE65-KITL) to drive KITL expression in photoreceptor cells and RPE cells, respectively, and then analyzed the effects of these two viruses on LIRD (see new Figure 5, Figure 5—figure supplement 1 and the response as described above in the Editor comment 1). Both viruses successfully drove overexpression of KITL in RPE and photoreceptor cells, respectively. Indeed, it is KITL from photoreceptor cells and not from the RPE that is protective for photoreceptor cells. The results are in the subsection “Overexpression of KITL in photoreceptor cells prevents light-induced retinal degeneration”.

4) The 661W cell line is a mouse retinoblastoma with features of multiple cell types and this is not an appropriate proxy for normal cells.

We appreciate your comment and understand your concerns. Please see the response to Editor comment 2 above.

5) The in vivo rescue was not well controlled. How many mice were used, how long were they followed, empty vector was not used in all experiments as should have been done.

We appreciate your suggestions and apologize for the lack of clarity. As mentioned above, for this revision, we used AAV8-CMV empty virus as a control virus and replaced all uninjected groups with AAV8-CMV empty virus injected control groups (Figure 4H-J; Figure 9; Figure 9—figure supplement 1). Regarding the in vivo rescue experiments, for *rd10/rd10*, we performed subretinal injection of AAV8-KITL virus into the eyes of such mice (n=5) at postnatal day 3 and examined their retinal functions at postnatal day 24 (subsection “KITL preserves photoreceptors and restores retinal function in genetic mouse models of retinal degeneration”). For *rd1/rd1* mice, the subretinal injection was performed (n=5) at postnatal day 1. The mice were kept under normal environmental condition for 20 days and then subjected to retinal degeneration analysis (figure legend of Figure 9—figure supplement 1). For LD treatment, the information can be found in the corresponding part of the Results section.

Reviewer #3:[…] I highly suggest giving the manuscript an additional editing pass for spelling and grammar as well as wording of some sentences (e.g. The sentence "We observed that after 2 weeks of injection, high-level KITL expression was restricted to photoreceptor cells and the RPE…" implies that the injections were performed for 2 weeks, when in reality they were done once and then the animals were followed up 2 weeks later. There are numerous occasions of this type of phrasing and other similar examples that are confusing and detract from the overall manuscript).

We appreciate your comments and are sorry for the confusion. We have replaced all sentences “after 2 weeks of injection” statements with “2 weeks after injection of the virus” throughout this revised manuscript.

I would also ask that the authors revisit their Discussion and explore more rationale as to the protective effect of KIT signaling on photoreceptors. The authors mainly restate their results and do not spend anytime analyzing how HMOX1 could be mediating its protective effect and this discussion is necessary to explain how KIT signaling can be protective. The authors also need to address alternative pathways downstream of KIT that may also be contributing to photoreceptor protection. The authors admit that HMOX1 is only capable of partially rescuing photoreceptors. Are there other pathways which may be of interest to examine next?

We appreciate your comments and apologize for the insufficient information. We have revisited our Discussion according to your suggestions and discussed the action model of KITL/KIT in photoreceptor cells, the potential manner by which HMOX1 could protect photoreceptor cells against LD, the regulation of relationship between the KITL/KIT and NRF2-HMOX1 axis and the limitation of this work (Discussion, fourth paragraph).

There are several ERG graphs that show only noisy traces which are not useful for assaying retinal function (Figure 3B, Kit+/+ no light damage photopic; Figure 8F, AAV8-KITL transduced rd10 homozygous animals; Figure 9—figure supplement 1B, both traces). If these panels are representative of all data collected for these experiments then they would either need to be repeated or omitted from the manuscript. The associated morphological data is sufficient to show a protective effect of KITL/damaging effect of removing KIT signaling without needing ERG data to show functional deficits.

According to your suggestions, we have repeated the ERG examination and replaced the previous images with improved ones (Figure 3B; Figure 9F; Figure 9—figure supplement 1B).

Is the AAV8 construct used for HMOX1 identical to the one used for KITL (besides the gene)? If so, why is there such broad expression of HMOX1 compared to KITL after transduction? The promoter used is CMV so one would expect broad expression in most cells of the retina.

We appreciate the reviewer’s comment. The basic vector of AAV8-HMOX1 contains a CMV promoter and is the same as that of AAV8-KITL. This AAV8-CMV vector has been reported to drive gene expression mainly in photoreceptor cells and RPE (Xiong et al., 2015). Our data also show that AAV8-KITL drives KITL expression mainly in photoreceptor cells and RPE. Similarly, we have repeated the examination of HMOX1 expression in the retinas infected by AAV8-HMOX1 and found that HMOX1 expression is mainly restricted to photoreceptor cells (see new Figure 8B in this revision).

KIT appears to be mainly expressed on the inner segments of photoreceptors while KITL appears to be mainly expressed on the outer segments of photoreceptors/RPE villi. Do photoreceptors normally express KITL or is this exogenously supplied by RPE during stress? Could the authors provide an in situ hybridization showing if KITL is expressed in photoreceptors or in RPE (or both)?

We appreciate the reviewer’s comment. According to your suggestions, we have examined KITL expression in RPE and photoreceptor cells of light-injured or uninjured retinas by gene expression analysis, including in situ hybridization (see new Figure 1I, Figure 1—figure supplement 2 and Figure 1—figure supplement 3). The data consistently show that photoreceptor cells upregulate *Kitl* expression in response to light stress (subsection “Light damage upregulates endogenous KITL in photoreceptor cells”, last paragraph). In addition, to analyze the cell specificity of the protective role of KITL, we used the photoreceptor cell-specific promoter RHO (AAV8-RHO-KITL) or the RPE cell-specific promoter RPE65 (AAV8-RPE65-KITL) to drive KITL expression in photoreceptor cells and RPE cells, respectively (see new Figure 5, Figure 5—figure supplement 1 and the response as described above in the Editor comment 1). Indeed, KITL exerts its protective role when expressed in photoreceptors and not when expressed in the RPE. The results are described in the subsection “Overexpression of KITL in photoreceptor cells prevents light-induced retinal degeneration”.